## ARTICLES

nature
# ecology & evolution
# Mass spectrometry of short peptides reveals common features of metazoan peptidergic neurons

Eisuke Hayakawa [1,9] ✉, Christine Guzman [1,9], Osamu Horiguchi [1,9], Chihiro Kawano[1], Akira Shiraishi [2], Kurato Mohri[1], Mei-Fang Lin [1,5], Ryotaro Nakamura[1], Ryo Nakamura [1], Erina Kawai [1,6], Shinya Komoto[1,7], Kei Jokura [3,8], Kogiku Shiba [3], Shuji Shigenobu [4], Honoo Satake [2], Kazuo Inaba [3] and Hiroshi Watanabe [1] ✉

**The evolutionary origins of neurons remain unknown. Although recent genome data of extant early-branching animals have shown that neural genes existed in the common ancestor of animals, the physiological and genetic properties of neurons in the early evolutionary phase are still unclear. Here, we performed a mass spectrometry-based comprehensive survey of short peptides from early-branching lineages Cnidaria, Porifera and Ctenophora. We identified a number of mature ctenophore neuropeptides that are expressed in neurons associated with sensory, muscular and digestive systems. The ctenophore peptides are stored in vesicles in cell bodies and neurites, suggesting volume transmission similar to that of cnidarian and bilaterian peptidergic systems. A comparison of genetic characteristics revealed that the peptide-expressing cells of Cnidaria and Ctenophora express the vast majority of genes that have pivotal roles in maturation, secretion and degradation of neuropeptides in Bilateria. Functional analysis of neuropeptides and prediction of receptors with machine learning demonstrated peptide regulation of a wide range of target effector cells, including cells of muscular systems. The striking parallels between the peptidergic neuronal properties of Cnidaria and Bilateria and those of Ctenophora, the most basal neuron-bearing animals, suggest a common evolutionary origin of metazoan peptidergic nervous systems.**

Understanding the origins of neurons remains one of the most intractable challenges in evolutionary biology. In the early phase of animal evolution, the transmission of cellular activity to specific target cells that are spatially separated in the macroscopic multicellular body should have an essential role in the behavioural and physiological responses of animals. However, it is still unclear which of the diverse physiological and genetic features of the complex nervous systems of extant animals are functionally deployed in ancestral neurons. There has been increasing interest in recent years in the extent to which the neural genes required for the functioning of the nervous system in Bilateria are present in the genomes of the basal metazoans Ctenophora, Porifera, Placozoa and Cnidaria[1–5]. Comparative genomic studies of the basal metazoans and their closest sister unicellular lineage Choanoflagellata have revealed that the origins of major neural components, including ion channels and synaptic proteins, predate the emergence of neurons[6–8]. The premetazoan origin of neural genes suggests that the functional neuron was established by a combination of protein modules with different evolutionary origins[6,9,10]. One of the key open questions is what genes or modules were deployed in the first functional neuron. To answer this, the nature of the neurons in the earliest divergent animal groups needs to be clarified. One of the extant groups is

Ctenophora (comb jellies), which is the most basal lineage bearing neuron-like cells with morphologically recognizable long processes (neurites) and presynaptic characteristics[11,12]. The genetic and physiological features of ctenophore neurons, however, largely remain to be experimentally examined. A deep single-cell-level inspection of neural genes in the ctenophore did not successfully provide a genetic signature characterizing the ctenophore neurons[13]. The lack of definitive neural markers in Ctenophora, together with the uncertainty of the phylogenetic position of this lineage[14], is regarded as support for the independent evolutionary origin of the ctenophore nervous system[8,11].

A key to identification and characterization of ctenophore neurons is identification of the neurotransmitters they use. Among neurotransmitter systems commonly functioning in bilaterians, the peptide system has been assumed to be related to the ancestral nervous system[15,16]. Neuropeptides are the most diverse group of neuromodulators, with multiple crucial roles in the neural control of behaviours in virtually all multicellular animals[16,17], and are thought to act as the major neuronal messengers in Cnidaria, the lineage that diverged before Bilateria (Fig. 1a)[18–20]. Genome-wide surveys have been carried out in the basal metazoans to describe the complete set of peptide genes[21]. However, classical peptide search methods

[1]Evolutionary Neurobiology Unit, Okinawa Institute of Science and Technology Graduate University, Okinawa, Japan. [2]Bioorganic Research Institute, Suntory Foundation for Life Sciences, Kyoto, Japan. [3]Shimoda Marine Research Center, University of Tsukuba, Shizuoka, Japan. [4]Center for the Development of New Model Organisms, National Institute for Basic Biology, Okazaki, Japan. [5]Present address: College of Marine Sciences, National Sun Yat-Sen University, Kaohsiung, Taiwan. [6]Present address: Marine Climate Change Unit, Okinawa Institute of Science and Technology Graduate University, Okinawa, Japan. [7]Present address: Imaging Section, Okinawa Institute of Science and Technology Graduate University, Okinawa, Japan. [8]Present address: Living Systems Institute, University of Exeter, Exeter, UK. [9]These authors contributed equally: Eisuke Hayakawa, Christine Guzman, Osamu Horiguchi. ✉e-mail: eisuke.hayakawa@oist.jp; Hiroshi.Watanabe@oist.jp

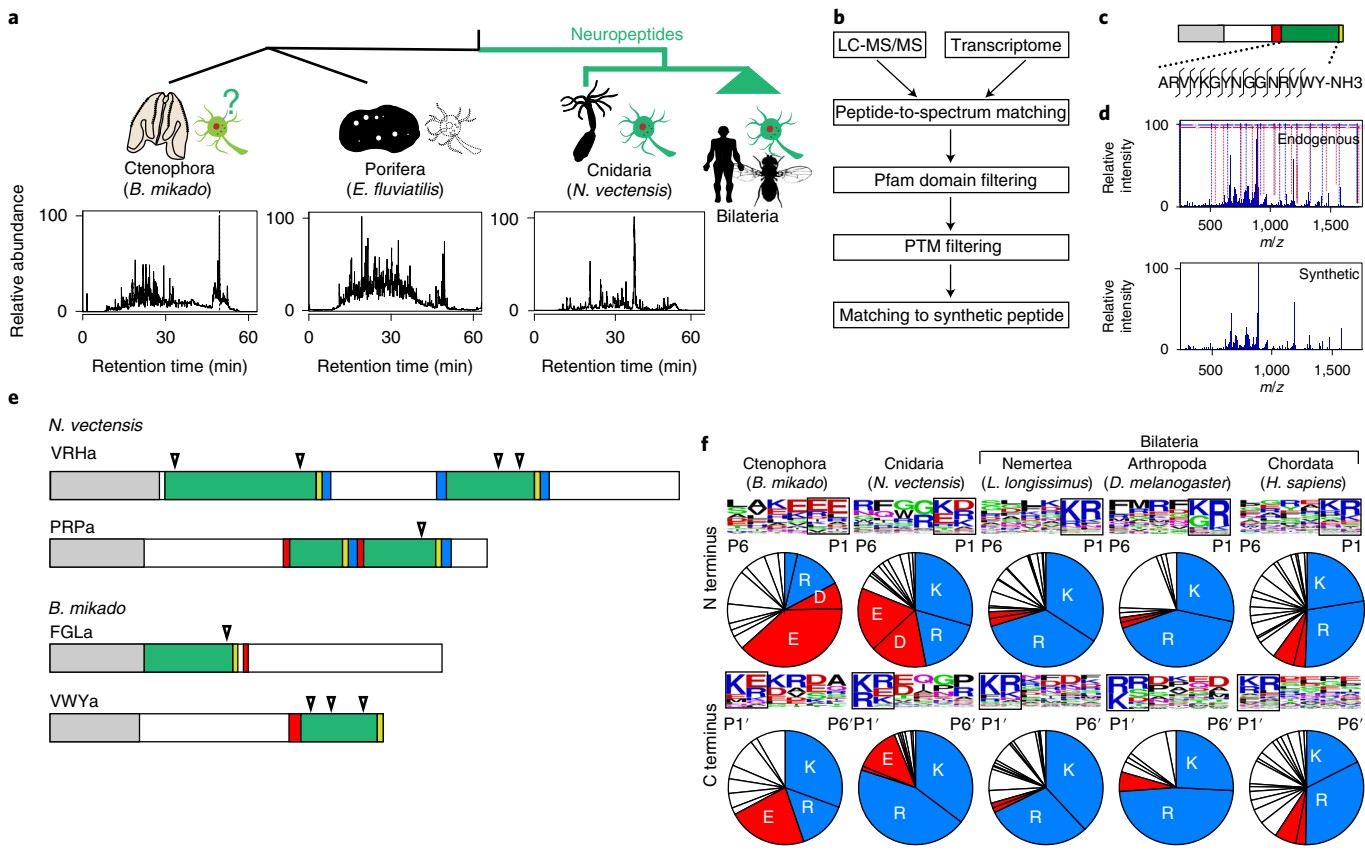

**Fig. 1 | Mass spectrometry-based identification of short amidated peptides in basal metazoans. a**, Phylogenic relationships of animals related to this study and known peptidergic systems are shown at the top. Total ion chromatograms of endogenous peptide fractions of basal metazoans *N. vectensis* (Cnidaria), *E. fluviatilis* (Porifera) and *B. mikado* (Ctenophora) are shown below. **b**, Data processing workflow of mass spectrometry-based neuropeptide identification. **c**, Schematic representation of the primary structure of the *B. mikado* VWYamide peptide. The positions of fragmentations observed in the fragment spectrum are indicated. **d**, Fragment spectra of endogenous and synthetic VWYamide. **e**, Schematic representation of neuropeptide precursors of FGLa, VWYa (*B. mikado*), PRPa and VRHa (*N. vectensis*). Grey, red, green and yellow boxes indicate signal peptide, cleavage site, mature neuropeptide and glycine as amide donor, respectively. Triangles denote the putative cleavage sites of neprilysin endopeptidase. No neuropeptide was detected from *E. fluviatilis*. **f**, Sequence logo map of N-terminal and C-terminal flanking regions and AA compositions of the cleavage sites (boxed) of neuropeptides in Metazoa. Basic (K and R) and acidic (D and E) AAs are shown in blue and red, respectively. Illustrations of fly, sea anemone and sponge are from phylopic.org.

## Results

**Basal metazoan peptides identified by mass spectrometry.** The experimental identification of ctenophore peptides is a necessary step to corroborate functional peptidergic (peptide expressing) cells and to clarify their physiological and genetic features. This is key to understanding the detailed characteristics of the ancestral nervous system and clarifying its origin(s), including whether it evolved multiple times. To obtain an experimentally validated overview of the mature structures and repertoire of peptides in early-branching metazoans, we performed mass spectrometry-based peptide identification for *Nematostella vectensis* (Cnidaria), *Ephydatia fluviatilis* (Porifera) and *Bolinopsis mikado* (Ctenophora), with a special focus on short peptides with amidated carboxyl termini, the major structural characteristic of neuropeptides among the metazoans tested[22]. We identified mature peptides encoded in 13 and 15 precursors

provided only a few candidate genes for short peptides in the genome of *Pleurobrachia bachei* (Ctenophora)[1]. Recently, a new search engine implemented with machine learning technology predicted a number of peptide gene candidates[12]. However, peptide prediction is still largely based on the structural features of bilaterian peptide precursors, which limits the reliability of in silico predictions especially in basal metazoans.

in *B. mikado* and *N. vectensis*, respectively (Fig. 1, Extended Data Fig. 1, Supplementary Tables 1 and 2 and Supplementary Data 1 and 2). In *E. fluviatilis* (Porifera), on the other hand, no amidated endogenous peptides were identified in this study. Furthermore, we did not detect in *B. mikado* the FMRFa-like and vasopressin-like amidated peptides that were predicted in *Pleurobrachia pileus* (Ctenophora)[23]. A cluster analysis of peptide precursor sequences did not detect similarity of ctenophore peptides to cnidarian and bilaterian peptides (Extended Data Fig. 2), as had been assumed in a previous work comparing peptide precursors among phylogenetically distantly related animal lineages[21]. None of the peptides related to 72 precursor candidates that have been predicted in the genome of ctenophore *P. bachei*[1] was found in *B. mikado* in our study. Recently, using the machine learning NeuroPID prediction platform, 30 neuron-enriched genes of *Mnemiopsis leidyi* (Ctenophora) were predicted as peptide precursor candidates[12]. Among them, we confirmed that the mature amidated short peptides NPWa, FGLa, RWFa, VWYa and WTGa were encoded by five genes, ML215411a, ML030511a, ML233326a, ML02736a and ML02212a, respectively. Other peptide genes identified in this study, NVFa/NIFa1 (ML40299a), NVFa/NIFa2 (ML16906a), TFQa (ML030215a), LNSa (ML218828a) and PARa (ML1541126a), have not been predicted elsewhere (Supplementary Table 3). The mass spectrometry-based

structural identification of mature peptides provided information about their cleavage sites and modifications and enabled us to examine the processing mechanism of the precursors, generate antibodies and perform biological assays using the mature peptides. In addition to basic (K and R) cleavage sites, which are commonly used for cleavage of neuropeptide precursors in bilaterians, we found that acidic (D and E) residues were also major cleavage sites both in *N. vectensis* and *B. mikado* (Fig. 1f). Recently, acidic cleavage sites have also been recognized in bilaterians[24], which may indicate the deep evolutionary roots of this underappreciated system. The structures of mature neuropeptides also enabled us to clarify evolutionarily conserved features, including amino-terminal pyroglutamation for peptide stability, and to detect putative target sites for neprilysin endopeptidase (Fig. 1e and Extended Data Fig. 1), which can rapidly degrade and inactivate secreted neuropeptides[25]. As neuropeptides are generally not recycled back into the neuron after secretion, their enzymatic inactivation is important to facilitate neuronal regulation of target cell responses. These data demonstrate that the biosynthetic and degradation mechanisms of neuropeptides are conserved among Ctenophora, Cnidaria and Bilateria.

**Morphology of ctenophore peptide-expressing cells.** We next produced antibodies against the amidated peptides NPWa, VWYa, FGLa, WTGa, RWFa and GVFa (a peptide contained in the NVFa precursor) of *B. mikado*, and against RFa, PRGa, QWa and HIRa of *N. vectensis*, and analysed the morphology and localization of the peptide-expressing cells (Fig. 2 and Extended Data Fig. 3). Immunostaining of *B. mikado* cydippid larvae using NPWa, VWYa and WTGa antibodies enabled us to visualize cells with cell processes (neurites), indicating that these amidated short peptides are functional in neurons (Fig. 2a–c). Our staining of these peptides, which we term neuropeptides, also demonstrated the complex neuronal architecture of *B. mikado*. NPWa, VWYa and WTGa were expressed in neurons constituting the subepithelial neural network (SNN) (Fig. 2g–i). VWYa-expressing (VWYa⁺) neurons extended multiple neurites and formed part of the polygonal SNN structure, confirming that the tubulin-positive network structure in ctenophore species consists of neurons (Fig. 2w–w″)[26]. In addition to the SNN, neurons expressing these neuropeptides developed the nerve plexus beneath each comb row in *B. mikado*. The striking morphological difference among these neuropeptides was that NPWa⁺ neurite bundles connected the apical organ and each comb row of the nervous system along aboral ciliated grooves (Fig. 2j–l).

As well as in neurons, we detected immunofluorescence signals of peptides in neurite-less cells of *B. mikado*. The neuropeptides (NPWa, VWYa and WTGa) and other short amidated peptides (FGLa, RWFa and GVFa) were expressed in cells with no visible neurite at the apical organ, tentacles or pharynx (Fig. 2m–v), suggesting that they have neuroendocrine functions. The expression of mature VWYa and WTGa in neurons of the SNN, comb row, apical organ and tentacles; of FGLa in the apical organ and pharynx; and of RWFa in tentacles of *B. mikado* was consistent with their messenger RNA (mRNA) distributions in *M. leidyi*[12]. This was not surprising, because both *M. leidyi* and *B. mikado* belong to the order Lobata and are members of the genera most closely related to each other[27]. To obtain a wider view of the anatomical and physiological features of ctenophore nervous systems, we performed immunostaining of adult combless benthic ctenophore *Vallicula multiformis* (order Platyctenida) using antibodies against NPWa, VWYa, FGLa and WTGa (peptides that have conserved C-terminal sequences in these species). We confirmed that a neural network of VWYa⁺ neurons also existed in this benthic ctenophore (Extended Data Fig. 4), demonstrating the generality of the VWYa⁺ SNN structure in the phylum Ctenophora. On the other hand, we observed differences in morphology and localization of peptidergic cells between free-swimming lobate and benthic species. In *V. multiformis*, NPWa⁺

cells did not extend neurites, and their cell bodies were closely associated with VWYa⁺ neurons. FGLa⁺ cells were not detected at the apical organ but were distributed at the body margin. In the SNN of *V. multiformis*, some VWYa⁺ neuronal cell bodies bore actin-rich protrusions, suggesting putative sensory functions.

**Conserved molecular machinery in metazoan peptidergic cells.** To further examine the nature of basal metazoan neurons at the molecular level, we used single-cell transcriptomic data from *N. vectensis*[28] and *M. leidyi*[13] and compared the genetic signatures of peptide-expressing cells, with a special focus on the cell clusters designated as neurons (*N. vectensis*) or unknown (*M. leidyi*). The neuronal clusters of *N. vectensis* were divided into peptidergic and uncharacterized neurons (Extended Data Fig. 5). For *M. leidyi*, we identified peptidergic cell clusters (C27, C29–35, C40 and C55) (Fig. 3) that partially overlapped with putative neuropeptide-expressing cells that were recently reported[12]. Among these cell clusters, four (C33, C34, C35 and C40) expressed the VWYa, NPWa and WTGa neuropeptides. Immunostaining confirmed that these peptides were expressed in cells including neurite-bearing neurons; therefore, these four cell clusters consist at least in part of peptidergic neurons. A comprehensive survey of protein components responsible for functions of neuropeptide-expressing neurons in *Bilateria*[17,29] showed a rich repertoire of gene homologues for neuropeptide signalling molecules with different expression levels and patterns (Supplementary Data 3). To gain more insight into the function of neuropeptide-expressing cells, we focused on the highly expressed homologues in each family. As shown in Fig. 3 and Extended Data Fig. 5, the highly expressed homologues that were essential for peptidergic function in Bilateria were also found in the peptidergic clusters in both *N. vectensis* and *M. leidyi*. It was clear that a set of proteases including carboxypeptidases, peptidylglycine α-amidating monooxygenase (PAM) and neprilysin, which are required for synthesis and degradation of active neuropeptides, were abundantly expressed. Among these enzymes, PAMs are of particular interest as they are critical for peptide amidation and have been identified in metazoans including Cnidaria and Porifera[30]. Our structural and phylogenetic analyses of ctenophore PAM proteins demonstrated that ctenophore PAM genes harbour domains required for peptide amidation (Supplementary Figs. 1 and 2). Ca²⁺ and K⁺ channels and a number of genes essential for regulated secretion of peptide-containing vesicles were also functional in the peptidergic clusters. The enrichment of genes encoding proteins involved in biosynthesis and quenching of neuropeptides and in regulation of vesicular secretion indicates that peptides function as signalling molecules in these cell clusters. Immunostaining of neuropeptides also enabled us to visualize the broad intracellular distribution of peptide-containing vesicles, probably dense-core vesicles (DCVs), which have been identified in *M. leidyi*[12] (Fig. 2g–i). DCVs store neuropeptides and are distributed in neuronal cell bodies and neurites in neuroendocrine cells in Bilateria[31] and are known to be involved in non-synaptic volume transmission (extrasynaptic transmitter release that activates cells at a distance)[32]. Neurexin, a core presynaptic membrane protein coordinating synaptic transmission, and PDZ domain-containing postsynaptic proteins were not abundantly expressed in the peptidergic cells, but they were enriched in the other cell clusters of *M. leidyi* (Fig. 3). The broad spatial arrangement of neuropeptide-positive vesicles in *B. mikado* neurons is reminiscent of that found in neuroendocrine cells in Bilateria, suggesting that peptidergic signalling in Ctenophora is mediated by volume transmission. We observed a similar expression signature of transcription factors (TFs) to that found for key molecules in peptide signalling. Highly expressed homologues of TF families, including bHLHA, SoxB, and Lim- and Paired (PRD)-class homeobox genes, were enriched in the peptidergic clusters of the basal metazoans (Fig. 3 and Extended Data Fig. 5). Notably, in ctenophore species

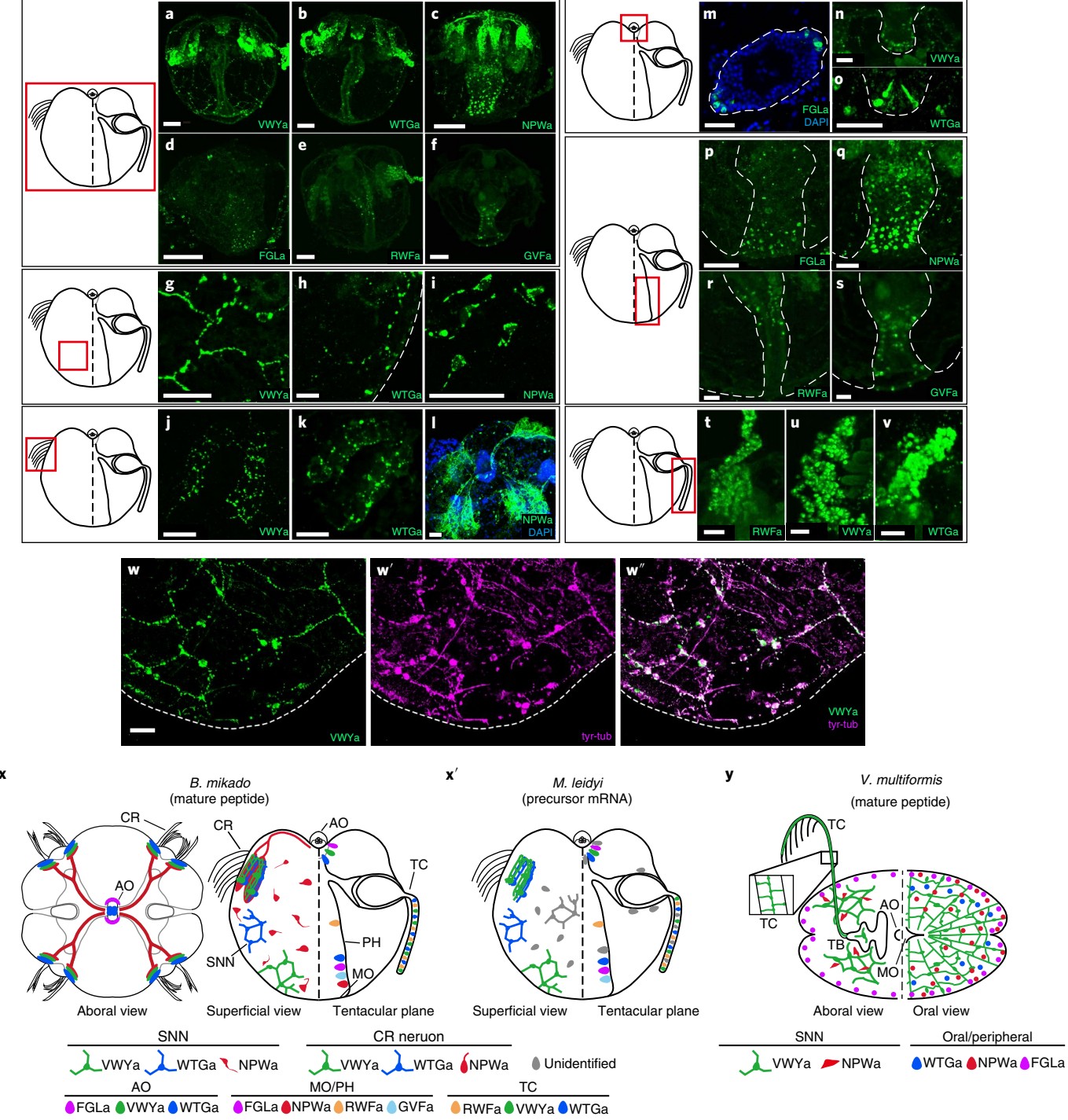

**Fig. 2 | Morphologies and distributions of ctenophore peptide-expressing cells.** The staining shown in each row focuses on the area enclosed by the red square in the adjacent schematic diagram. **a–f**, Side views of *B. mikado* larvae stained with antibodies against VYWa (**a**), WTGa (**b**), NPWa (**c**), FGLa (**d**), RWFa (**e**) and GVFa (**f**). **g–i**, High-magnification view of the SNN consisting of VWYa⁺ (**g**), WTGa⁺ (**h**) and NPWa⁺ (**i**) neurons. The dotted line in **h** indicates the outline of the larval body. **j–l**, High-magnification view of VWYa⁺ (**j**), WTGa⁺ (**k**) and NPWa⁺ (**l**) neurons at the nerve plexuses beneath comb rows. **m**, Aboral view of the apical organ stained by anti-FGLa antibody. The dotted line indicates the outline of the apical organ. **n,o**, VWYa⁺ (**n**) and WTGa⁺ (**o**) cells at the epithelial floor of the apical organ. The dotted line indicates the outline of the epithelial floor. **p–s**, Distribution of FGLa⁺ (**p**) NPWa⁺ (**q**), RWFa⁺ (**r**) and GVFa⁺ (**s**) cells in the pharynx. **t–v**, Expression of RWFa (**t**), VWYa (**u**) and WTGa (**v**) in the tentacle. **w–w′′**, Staining of VWYa⁺ neural network (green, **w**) and tyrosinated tubulin (tyr-tub; magenta, **w′**) and costaining of both (**w′′**). **x,x′**, Schematic diagram of the spatial distribution of amidated-peptide-expressing neurons and cells in *B. mikado* larva (**x**) and precursor-mRNA-positive neurons and cells in *M. leidyi* (**x′**). The left and right sides of the side view show a superficial view and the tentacular plane, respectively. In **x**, the aboral view is also shown on the left. **y**, Schematic diagram of the spatial distribution of amidated-peptide-expressing neurons and cells in *V. multiformis* adults. The left and right show the aboral side and oral side, respectively. Neurons with neurites are represented by circles with lines, and neurosecretory-like cells are represented by ovals. AO, apical organ; CR, comb row; MO, mouth; PH, pharynx; TB, tentacle bulb; TC, tentacle. Scale bar, 50 μm (**a–f**) or 20 μm (others). Pseudo-colours in **a–w** were applied using ImageJ software.

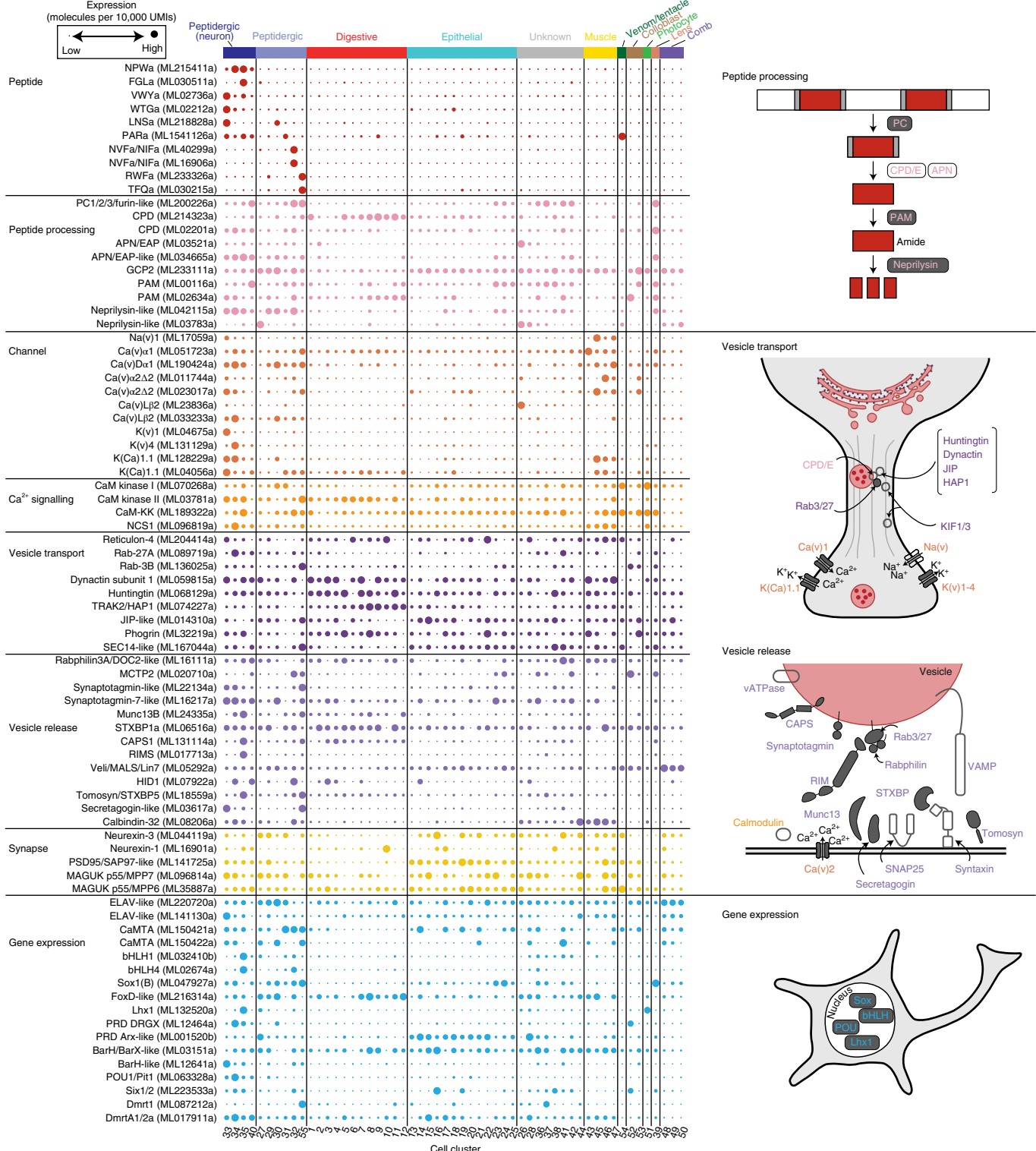

**Fig. 3 | Genetic signature of peptidergic cells in the ctenophore *M. leidyi*.** Expression of gene homologues involved in peptidergic neuronal function across the different cell clusters of *M. leidyi* based on scRNA-seq data[12]. The dot plot shows the normalized expression values (molecules per 10,000 UMIs) scaled by gene. Dot size from smallest to largest corresponds to lowest and highest expression, respectively. The genes with highest expression in each gene family were selected (Methods). Dots are coloured according to gene functional categories. Schematic representations of the major functional modules required for peptidergic signalling are shown on the right side. The molecules abundantly expressed in the peptidergic clusters are shown in grey.

*P. pileus* and *M. leidyi*, the SoxB, Lhx1, PRD homeobox genes have been demonstrated to be expressed abundantly in the apical organ, comb row and pharynx[33–36], where peptidergic cells develop

(Fig. 2j–s). These results indicate that the *M. leidyi* nervous system has a highly peptidergic nature and is equipped with protein machinery in common with the peptidergic nervous systems of

Cnidaria and Bilateria. In addition, major TF groups that have evolutionarily conserved roles in neurogenesis and neural cell type specification in Bilateria and Cnidaria were expressed in the peptidergic cells of *M. leidyi*, suggesting that a similar transcriptional regulatory programme may dictate neural development in Ctenophora. In addition to elucidating the peptidergic nature of the neurons in *M. leidyi*, we examined whether they had glutamatergic ability, as glutamate has been proposed as a candidate chemical neurotransmitter[1,9]. In *M. leidyi*, we found that homologues of the solute carrier family 17 (SLC17) vesicular transporters, which are required for the accumulation of glutamate in secretory vesicles, were not enriched in the peptidergic clusters but were abundantly expressed in other cell clusters (Extended Data Fig. 6). The most abundant SLC17 homologue (ML011726a) in *M. leidyi* was expressed exclusively in digestive cell clusters, in sharp contrast to the rich neuronal expression of SLC17 genes in *N. vectensis*. ML21903a, the next most abundant SLC17 homologue, showed broader expression in several cell cluster categories, including peptidergic clusters. This suggests that some of the peptidergic cells have the ability to store glutamate in secretory vesicles. Consistent with this, translucent vesicles with a diameter of about 50 nm have been reported to have been observed in some peptidergic cells in the vicinity of presynaptic triads of ctenophores[12]. However, it remains unknown whether these translucent synaptic vesicle-like entities are secretory vesicles and, if so, what chemicals they contain. In *M. leidyi*, a number of ionotropic and metabotropic glutamate receptors were expressed in neurons, indicating that the neurons are able to receive glutamate signals. The function of glutamate in ctenophores and its involvement in their nervous systems needs to be examined in the future.

**Peptide functions and putative receptor distribution.** To elucidate the biological roles of neuropeptides in the ctenophore nervous system, we next performed functional screening of the neuropeptides using cydippid larvae of *B. mikado*. Treatment of the larvae with the synthesized mature NPWa peptide (IGSDIKLVPGAGGNPWa) but not its reverse peptide (control) induced adradial canal contraction and mouth opening (Fig. 4a–c and Extended Data Fig. 7). This is consistent with the localization of the NPWa+ nervous system at the aboral subepithelium and neuroendocrine cells around the mouth. The VWYa neuropeptide (ARVYKGYNGGNRVWYa), on the other hand, triggered expansion of the oral epithelium, where VWYa+ neurons form the subepithelial network, resulting in an increase in body volume (Fig. 4d,e). The effects of ctenophore neuropeptides on muscular contractile responses were in line with peptide regulation of body contraction in Placozoa and Cnidaria[19]. To gain more insight into the categories of peptide target cells, we searched for putative peptide receptors of *M. leidyi* using the peptide-descriptor (PD)-incorporated support vector machine (SVM), a machine-learning-based method for prediction of peptide–G-protein coupled receptor (GPCR) pairs[37,38] (Supplementary Table 4). A total 122 pairs of 11 peptides and 38 GPCRs (among 538 GPCRs) were predicted to have ligand–receptor relationships (Fig. 4f and Supplementary Figs. 3, 4 and 5). Although the functions of these GPCR genes are yet to be validated empirically, the prediction shows interesting trends of peptide target cell types and suggests high-level complexity of neuronal function in *M. leidyi* (Fig. 4f,g). First, many of the GPCR genes were expressed in multiple cell clusters or categories; second, GPCR candidates as receptors for WTGa and FGLa were both expressed in putative sensory cells of the apical organ, showing rich expression in epithelial clusters; and, last, the majority of target GPCRs for VWYa were expressed in peptidergic clusters, consistent with the colocalization of VWYa+ neurons with other neurons (Fig. 2 and Extended Data Fig. 4). The complex expression pattern of the GPCR peptide receptor candidates, together with the diverse distributions of neurite-less neurons in cnidarians and ctenophores, implies that the primary mode of neuronal

transmission in the basal metazoans is organized mainly via the specificity of peptide–receptor pairs, as was recently proposed[17].

## Discussion

In this study, we identified a number of short amidated peptides in *B. mikado* and *N. vectensis*. Unlike sequence-based prediction, mass spectrometry can be used to identify structures of new peptides without prior knowledge, such as cleavage sites and conserved structures. On the other hand, it requires samples with high concentrations of peptides, and precursor genes need to be present in a transcriptome data set. Although putative precursors for amidated peptide phonexin in *M. leidyi* and some poriferan species have recently been predicted by homology search[39], we did not detect any transcript of phonexin in *B. mikado* or *E. fluviatilis*. We assume that there are amidated and non-amidated peptide species that are yet to be discovered. Taking advantage of the structural information of peptides we have identified here, we expect future studies using homology search or machine-learning-based prediction to reveal even more peptides in basal metazoans. A more comprehensive list of short peptides in basal metazoans will greatly contribute to our understanding of the evolutionary processes underlying their diversity and functions.

Immunostaining of short peptides demonstrated a high degree of variation in the morphology and localization of peptide-expressing neurons and cells in *B. mikado* and *V. multiformis*. In developing *N. vectensis*, short-peptide-expressing cells developed a well-defined neural network at the primary polyp stage, whereas at the planula stage, they appeared to be neuroendocrine-like cells without long neurites (Extended Data Fig. 3). The morphological diversity of peptide-expressing cells may indicate flexibility and/or plasticity of their functions in an early evolutionary phase of the peptidergic system. Taking these results together with recent findings that a neuroendocrine-like system regulates the behaviour of the other basal metazoan Placozoa[16,40], it is reasonable to postulate that the local neuroendocrine mode of signalling was responsible for behavioural control in ancestral Metazoa, whereas the network-dependent neural functionality represented secondary sophistication. In addition, given that glutamate was secreted mainly from non-neural cells in *M. leidyi* (Extended Data Fig. 6), it is reasonable to postulate that the glutamate transmitter system was already functional in non-neural systems, with secondary deployment as a neurotransmitter in the nervous system.

We found highly conserved machinery for peptide biosynthesis, release and degradation, which are essential for peptide signalling. The unexpected level of conservation of the genetic signature of peptide-expressing cells among metazoans is strongly suggestive of a common and single evolutionary origin of peptidergic cells. The localization of some of the identified peptides revealed by immunostaining cannot be explained by the cell cluster classification of *M. leidyi* that was recently proposed[12]. For example, VWYa and WTGa peptides were most highly expressed in C33, which was designated as the SNN in the previous work; however, strong signals for these neuropeptides were also detected in cells other than SNN in *B. mikado*. This discrepancy could be attributed to differences in the neural organization of the two species. Alternatively, in *M. leidyi*, it may have been caused by the association of the adult single-cell transcriptome with larval neural staining. Therefore, in-depth analysis—possibly involving costaining of cluster-specific neural genes, as has been suggested recently—is needed to achieve detailed characterization of neural (sub)types by single-cell transcriptomics[41].

The origin of neurons has long been debated in view of the various characteristics of neurons (for example, morphological, physiological and chemical features). It has been proposed that neurons evolved independently from secretory cell types multiple times, given the huge diversity of neuronal physiological characteristics, including chemical transmitter composition[42]. However, the chemical substances that are generally called 'neurotransmitters'

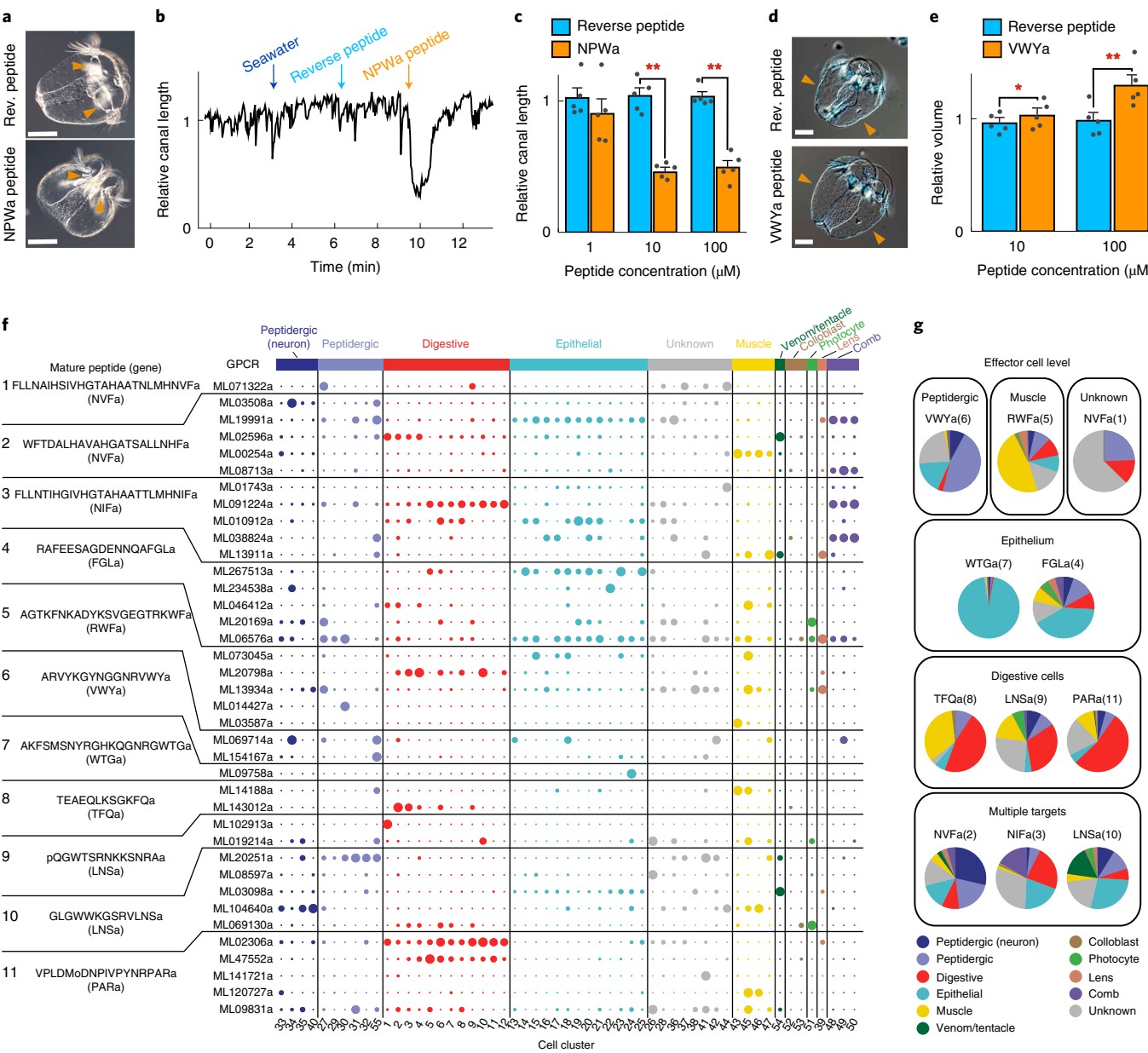

**Fig. 4 | Functional characterization and receptor prediction of ctenophore neuropeptides. a**, Synthetic NPWa neuropeptide (IGSDIKLVPGAGGNPWa) induced contraction of the adradial canal (orange arrowheads) and mouth opening of *B. mikado* larvae. A peptide with the reverse AA sequence was used as a negative control. **b**, Typical change in the relative length of the adradial canal following treatment with seawater, 10 μM reverse peptide and 10 μM NPWa peptide. The time points at which the changes of medium took place are indicated by blue (seawater), light blue (reverse peptide) and orange (NPWa peptide) arrows. **c**, Quantitative analysis of the effect of increasing peptide concentrations on adradial canal contraction (*n* = 5 biologically independent animals for each concentration). Error bars represent the s.e.m. (**P < 0.01). **d**, Synthetic VWYa neuropeptide (ARVYKGYNGGNRVWYa) induced expansion of the epithelium (orange arrowheads). **e**, Quantitative analysis of the effect of VWYa on body volume (*n* = 5 biologically independent animals for each concentration). Error bars represent the s.e.m. (*P < 0.05, **P < 0.01). **f**, Cell cluster distribution of putative peptide GPCR expression. The representative peptide sequences from each peptide gene (in parentheses) in *M. leidyi* are shown on the left. The dot plot shows the normalized expression values (molecules per 10,000 UMIs) scaled by gene. Dot size from smallest to largest corresponds to lowest and highest expression, respectively. Dots are coloured according to the cluster category. **g**, Cell targets of representative peptides based on the expression of their predicted GPCRs (Methods). Scale bars, 50 μm (**a,d**). Statistics for **c**: two-tailed paired *t* test (1 μM: *t* = 2.448, d.f. = 4, *P* = 0.07059, 95% CI = −0.01615414, 0.25699004; 10 μM: *t* = 11.521, d.f. = 4, *P* = 0.0003241, 95% CI = 0.4416221, 0.7220451; 100 μM: *t* = 8.6694, d.f. = 4, *P* = 0.0009742, 95% CI = 0.3668352, 0.7125043); **P < 0.01. Statistics for **e**: two-tailed paired *t* test (10 μM: *t* = −3.0913, d.f. = 4, *P* = 0.03653, 95% CI = −0.4612434, −0.1623270; 100 μM: *t* = −5.792, d.f. = 4, *P* = 0.004417, 95% CI = −0.129532465, −0.006951329); *P < 0.05, **P < 0.01. Rev., reverse.

are widely used for intercellular signalling in non-neural cells in metazoans, as well as in plants and microorganisms. Therefore, when considering the most ancestral functionality of the neuron, it is unclear to what extent of the placement of particular chemicals may fulfil the 'neuronality' of the cell. More information about the Ctenophora nervous system is needed to further elucidate which features or combinations of features found in modern neurons can provide insight into the most ancestral state of neurons.

In this study, we demonstrated that the vast majority of the molecular components necessary for peptide processing and secretion are widely shared among neuronal and secretory cell clusters in *N. vectensis* and *M. leidyi* (Fig. 3 and Extended Data Fig. 5). It has been assumed that the C35 cell cluster is the only cell type homologous to cnidarian and bilaterian neurons[43]. Our analysis did not detect specific conserved characteristics that distinguished C35 from other neuronal clusters (Fig. 3). Instead, we also detected high expression of genes involved in peptide processing and vesicular transport and/or release in non-neural cell clusters categorized as digestive and epithelial. The presence of digestive enzymes and a secretory system should be a prerequisite for propeptide precursors to undergo processing and become mature short forms and to function under stimulus-responsive secretory control. If the peptidergic neurons function as the most ancestral system, the origin of neurons may presuppose the establishment of functional digestive cells and be evolutionarily related to their functional derivation. This idea has affinity with longstanding hypotheses proposing that the first neuron evolved from sensory–secretory cells[44–46]. We extend these hypotheses to envisage a 'digestive neuron', that is, a scenario in which the first neurons arose from digestive cells with sensory functions and begin to secrete and use proteolytic products of endogenous polypeptides. A digestive-system-related signalling system was probably indispensable for ancient multicellular heterotrophic organisms, enabling them to regulate digestion, behaviour and metabolism associated with feeding and to become active predators. Non-synaptic release or volume transmission is likely to be a dominant mode of peptide signalling in Ctenophora. Together with the high complexity of peptide ligands and their GPCRs, these findings suggest that neural function in the ancestral nervous system depended on the complexity and specificity of transmitter–receptor pairs rather than on physical constraints imposed by the targeted wiring of neurites.

## Methods

**Animal culture.** *N. vectensis* (Cnidaria) were cultured as previously described[47] with few modifications. Adult animals were kept at 18 °C and maintained at a salinity of one-third artificial seawater (35 g l$^{-1}$, pH 7.5–8.0, SeaLife Marine Tech) and fed with freshly hatched artemia two times per week. Spawning induction was performed at 26 °C under light for at least 11 h. *B. mikado* (Ctenophora) were cultured as previously published[48] with slight modifications. Adult individuals collected at Kamo Bay (Oki island, Shimane), Hakkeijima (Yokohama, Kanagawa), Nanao Bay (Nanao, Ishikawa) and Tabira Bay (Hirado, Nagasaki) in Japan were maintained in a 60 l aquarium with slow water circulation at Shimoda Marine Research Center (Shizuoka, Japan) and OIST Marine Science Station (Okinawa Japan). Individuals were fed artemia in the morning and evening, with two to three feedings of frozen copepod (Pacific Trading and Kyorin) in between. The adult stage of the animals was used for mass spectrometry analysis. Isolated and $H_2O_2$-treated gemmules of *E. fluviatilis* (Porifera) were provided by Prof. Noriko Funayama (Kyoto University). Gemmules were cultured in M-medium (1 mM $CaCl_2$, 0.5 mM $MgSO_4$, 0.5 mM $NaHCO_3$, 0.05 mM KCl and 0.25 mM $Na_2SiO_3$) at 23–24 °C. To obtain large colonies for mass spectrometry analysis, we cultured the newly hatched juvenile sponges (~1 mm in diameter) with phytoplankton (50 μl l$^{-1}$ freshwater *Chlorella* sp. (Nikkai Center) and 1 ml l$^{-1}$ PhytoChrome (Brightwell Aquatics)) for 2–3 months. *V. multiformis* (Ctenophora) were collected at Kiyoshi-Hiroshi sea-grape farm in Ginoza (Okinawa, Japan). They were maintained in 1.5 l seawater and fed with freshly hatched artemia two times per week.

**Mass spectrometry-based peptide identification.** Endogenous peptide fractions of *N. vectensis*, *E. fluviatilis* and *B. mikado* were extracted by homogenization in acidified methanol solution (90% methanol, 9% ultrapure water and 1% formic acid) with a bead homogenizer (Bead Smash 12, Waken B Tech), followed by removal of debris and large proteins by centrifugation (4 °C at 13,000g for 20 min). The supernatants were vacuum dried with a GeneVac EZ-2 Elite (SP Scientific) and then subjected to solid-phase extraction with an Oasis HLB 1 cc cartridge (Waters). The eluents were vacuum dried and stored at −80 °C for liquid chromatography-coupled tandem mass spectrometry (LC-MS/MS) analysis. The sample preparation procedure was performed independently four times for each animal species, and the resulting samples were subjected to the following analysis separately. For each animal species, one of the four extract replicates was subjected to fractionation to reduce sample complexity before LC-MS/MS analysis. Fractionation was performed according to the high-pH reversed-phase protocol[49]

with slight modification. The details of peptide extraction, solid-phase extraction and fractionation are described in the Supplementary Information.

The peptide samples from each animal species were analysed by LC-MS/MS analysis as follows. Dried samples were resuspended and analysed with a nanoflow liquid chromatography system (nanoACQUITY, Waters) coupled with an Orbitrap mass spectrometer (Fusion Lumos, Thermo Fisher Scientific). The samples were loaded using an autosampler into a trap column (nanoACQUITY UPLC 2G-V/M Symmetry C18, 5 μm, 180 μm × 20 mm, Waters) and subsequently into an analytical column (nanoACQUITY UPLC HSS T3, 1.8 μm, 75 μm × 150 mm, Waters). Separation of peptides was performed with a 65 min gradient (mobile phase A, 0.1% formic acid in water; mobile phase B, 0.1% formic acid in acetonitrile) with a flow rate of 500 nl min$^{-1}$. Fragment ion spectra of peptides were acquired by a data-dependent approach by using CHarge Ordered Parallel Ion aNalysis with slight modification[50]. Details of the parameter settings used for mass spectrometry, the liquid chromatography conditions, acquisition method and overall workflow are described in the Supplementary Information.

Peptide-to-spectrum matching was performed using PEAKS X software (PEAKS Studio v.10.0, Bioinformatics Solutions) and Mascot (v.2.7, Matrix Science). The raw data of the acquired MS/MS spectra were processed with PEAKS X software; PEAKS DB algorithms were used to perform peptide-to-spectrum matching. The search parameters were as follows: 5 ppm for precursor tolerance, 0.3 Da for fragment tolerance, no fixed modifications, oxidation of methionine (15.99491 Da), pyroglutamation from glutamic acid (−18.01057 Da) or glutamine (−17.02655 Da) and amidation at peptide C termini (−0.98402 Da). For Mascot, fragment ion spectra were extracted in mgf file format from the raw data using MS convert in ProteoWizard package 3.0.20139 (ref. [51]). The same search parameters for mass tolerance and variable posttranslational modification were used for Mascot. Searches were performed against amino acid (AA) sequences translated in six frames from the transcriptome data of the respective animal species. The results of peptide-to-spectrum matching were processed to select neuropeptide-related endogenous peptides as follows. HMMER[52] v.3.2.1 was used to scan precursor proteins of detected peptides for Pfam[53] (version 33.1) motifs. All peptides whose precursor protein had any Pfam motifs except for neuropeptide-related motifs (Supplementary Information) with an *e* value cutoff of $1 \times 10^{-10}$ were removed and not processed in the following analysis. In this study, we focused on naturally occurring peptides with C-terminal amidation as neuropeptide candidates. To this end, only peptides detected as C-terminally amidated forms were selected for further analysis. AA sequences of top scoring peptides in the filtered peptide-to-spectrum matching results were selected for peptide synthesis for further experimental validation. In cases where multiple peptides were found to be derived from the same precursor protein, one was chosen for synthesis. Peptide synthesis (~80% purity) was performed using ResPep SL (Intavis). The synthesized peptides were subjected to LC-MS/MS analysis using the same settings as described above. Fragment spectra acquired from the synthetic peptides were compared with the spectra of endogenously detected peptides for validation. Raw LC-MS/MS data of endogenous and synthetic peptides, peptide-to-spectrum matching results, AA sequences used for peptide-to-spectrum matching, raw output of search engines showing unfiltered lists of identified peptides with their precursors and other files related to peptide identification have been deposited in the ProteomeXchange Consortium via jPOST[54] with the data set identifier PXD030145. A complete list of identified peptides is provided in Supplementary Tables 1 and 2.

**Global cluster analysis of neuropeptide precursor similarity.** Similarities between metazoan neuropeptide precursors, including newly identified ones, were visualized using the same approach as Jékely[21]. AA sequences of neuropeptide precursors were assembled from various sources together with newly identified neuropeptide precursors (Supplementary Data 4). The sources and details of precursors used in this study are provided in the Supplementary Information. All neuropeptide precursor sequences were subjected to all-against-all Basic Local Alignment Search Tool (BLAST) search (blastp, v.2.11.0+)[55]. Clustering and visualization were performed using Cytoscape (v.3.5.1)[56] with an *e* value cutoff of $1 \times 10^{-5}$.

**Cleavage site analysis.** To assess actual cleavage sites in neuropeptide precursors in an unbiased way, we focused on neuropeptide structures that had been identified by mass spectrometry-based systematic peptide identification. AA sequences of neuropeptides were acquired from *Homo sapiens*, *Drosophila melanogaster* (Arthropoda) and *Lineus longissimus* (Nemertea). N-terminal and C-terminal cleavage sites (2 AA) and flanking regions (6 AA) of the collected neuropeptides in their precursors were extracted and subjected to composition analysis or sequence logo analysis with WebLogo[57]. The sources of peptide information for each animal species and the details of the cleavage site analysis are described in the Supplementary Information.

**Transcriptome analysis.** Total RNA of *B. mikado* was extracted from whole bodies of adult individuals collected from Hakkeijima (Yokohama, Kanagawa), Japan. RNA sequencing was performed using a HiSeq 2500 (Illumina) with 150-bp paired-end reads in two lanes. A total of 157.1 million reads were obtained and trimmed using libngs (https://github.com/sylvainforet/libngs) with minimum quality of 25 and minimum size of 70 nucleotides. High-quality paired reads were

retained for de novo assembly with Trinity[58] v.2.3.2. Reads from different locations were assembled independently.

**Phylogenetic analysis.** Sequences of PAM genes were collected from GenBank, except in the case of *B. mikado*, for which sequences were identified from transcriptome data generated by Jokura et al.[48]. The identification of PAM sequences was performed by blastp search with default *e* value cutoff, followed by protein classification based on RPS-BLAST against the NCBI Conserved Domain Database[59]. For phylogenetic tree reconstruction, PAM sequences containing both PHM and PAL domains were aligned by ClustalW[60], with careful manual examination before trimming. The alignment of PAM is available on request. Owing to uncertainties regarding the functions, causes and origins of the incomplete PAM homologues, 'short' gene models were not included in the phylogenetic analyses. The best substitution model was estimated by SMS[61] as WAG+I+G+F. Phylogenetic trees were built using PhyML[62] v.3.0 for maximum-likelihood analysis and MrBayes[63] v.3.2.3 for Bayesian inference. For maximum-likelihood analysis, 100 bootstraps were calculated. For Bayesian inference, two runs (four chains) each of 10 million generations were calculated, with the initial 25% discarded as burn-in. The average standard deviation of split frequencies between runs converged to less than 0.002, and the average potential scale reduction factor was 1.000.

**Gene expression profiling analysis.** Previously generated single-cell RNA sequencing (scRNA-seq) data of adult *N. vectensis* and *M. leidyi* were used for the gene expression analysis[13,28]. Raw unique molecule identifier (UMI) counts were retrieved from http://compgenomics.weizmann.ac.il/tanay/?page_id=724. A custom Python script was written to generate gene–cell-cluster expression matrices. Cell clusters were assigned a cell type identity according to the original papers, which generated the scRNA-seq data, with few modifications. In *M. leidyi*, clusters 52 and 53 were renamed as colloblasts and cluster 54 as tentacle/venom[64]. A global-scaling normalization method was used to normalize the gene expression measurements for each cell by the total expression, multiplied by a scale factor (1,000 for *N. vectensis*; 10,000 for *M. leidyi*). Cells with total UMI counts of less than 10 were removed from the analysis. *N. vectensis*[65,66] and *M. leidyi*[2] protein models were annotated by blastp alignment to the *Caenorhabditis elegans* and human UniProt/NCBInr databases with an *e* value cutoff of $1 \times 10^{-5}$. We also performed reciprocal blastp against the *N. vectensis* and *M. leidyi* databases using *C. elegans* DCV proteins as queries. All putative homologues were manually queried with the NCBI BLAST webpage to further confirm high-quality alignment and identity. Lists of annotated genes are provided in Supplementary Tables 5 and 6. To determine the genetic signatures of peptide-positive clusters, we focused our analysis on the homologues with the highest expression, which are more likely to have a greater effect on cell function. On the other hand, low-expressed homologues often show greater cell type specificity, but they are more susceptible to noise during scRNA-seq. We also excluded genes for which high expression was observed in lineage-specific cell types (for example, cnidocytes of *N. vectensis* and colloblasts of *M. leidyi*). The code used is available at https://github.com/oist/scrna-counts.

**Antibody production.** The C-terminal structures of amidated peptides (Cys-NPWamide, Cys-MHGVFamide, Cys-QAFGLamide, Cys-GTRRWFamide, Cys-NRVWYamide and Cys-RGWTGamide for *B. mikado*; and Cys-HIRamide, Cys-PRGamide, Cys-QWamide and Cys-RFamide for *N. vectensis*) were used to immunize rabbits with Freund's Complete Adjuvant for the first injection or Freund's Incomplete Adjuvant for boosting (Scrum Inc.). After checking the immunoreactivity by ELISA, each antiserum was affinity purified using peptide-conjugated beads.

**Immunofluorescence staining.** Immunostaining for *N. vectensis* planula larvae (4 days postfertilization) and juvenile polyps (10 days postfertilization) were performed as previously described[47] with slight modifications. Before fixation, polyps were relaxed by adding 0.3 M $MgCl_2$ solution dropwise. Animals were fixed with Zamboni's fixative (0.2% picric acid, 2% paraformaldehyde (PFA) in phosphate-buffered saline (PBS), 0.1 M phosphate buffer 7.2) overnight at 4 °C. After washing three times with PBS-Tween-20 (0.2%), specimens were incubated in 0.1 M glycine (pH 7.0) in PBS for 15 min at room temperature followed by washing twice with PBS-Triton (0.2%). The specimens were then incubated in blocking solution (10 mg ml⁻¹ bovine serum albumin (BSA), 5% normal goat serum, 0.1% sodium azide in PBS-Tween-20 (0.2%)) for 1 h at room temperature and then incubated with the anti-neuropeptide primary antibody (1:200 in blocking solution) overnight at 4 °C. The specimens were washed twice with PBS-Triton (0.2%) and then incubated with the Alexa-488-conjugated anti-rabbit secondary antibody (1:500 in blocking solution) (111-545-003, Jackson Immuno Research Laboratories) for 1 h at room temperature. The specimens were washed three times with PBS-Triton (0.2%) and mounted on a slide glass with SlowFade antifade reagent (S36937, Thermo Fisher Scientific) under a cover glass. Specimens were imaged at multiple planes covering an entire *z* axis using a fluorescence microscope (ECLIPSE Ni, Nikon) equipped with a Ds-Ri2 (Nikon). Acquired cross-sections at different depths were mathematically deblurred using a blind

deconvolution algorithm with 30 iterations (NIS-Elements AR v.5.02.03, Nikon). Deconvolved sections were then stacked for maximum intensity projection.

Immunostaining for *B. mikado* larvae (1–2 days old) was performed according to the method of Pang and Martindale[35] with slight modifications. Specimens were collected in a 3-cm plastic dish. An equal volume of 6.5% MgCl₂ solution was added and specimens were left to rest for 15–20 min until the tentacles extended. Larvae were transferred to a fixation buffer (4% PFA, 0.05% glutaraldehyde, 0.5 M NaCl, 0.1 M 3-(N-morpholino)propanesulfonic acid) and fixed for 1 h at 4 °C. Samples were washed with PBS containing 0.05% Tween-20 (five times for 5 min each time). Samples were transferred to a blocking solution (10% normal goat serum in PBS), and blocking was performed for 1 h at room temperature. After blocking, larvae were transferred to the primary antibody solution (1:500 dilution with blocking solution) and reacted at 4 °C overnight. Samples were washed with PBS containing 0.05% Tween-20 (six times for 15 min each time) then incubated with Alexa-488-conjugated anti-rabbit secondary antibody (1:500, A-11008, Thermo Fisher Scientific) for 3 h at room temperature. Samples were washed with PBS containing 0.05% Tween-20 (three times for 15 min each time), mounted in PBS containing 0.05% Tween-20 and observed by confocal microscopy (Olympus Fluoview FV10i).

Immunostaining for *V. multiformis* was performed as for *B. mikado* with slight modifications. Specimens were placed on cover glasses in culture wells filled with seawater. After animals had adhered to the glass, seawater was removed and the animals were fixed with chilled 4% PFA in PBS at 4 °C overnight. After washing with PBS containing 0.1% Triton X100 for 15 min three times at room temperature and blocking with 1% BSA in PBS for 2 h at room temperature, samples were incubated with primary antibody at 4 °C for overnight in 1% BSA in PBS using the following dilutions: anti-VWYa, 1:100; anti-NPWa, 1:250; anti-FGLa, 1:300; anti-WTGa, 1:250 and anti-tyrosinated tubulin 1:500 (T9028, Sigma-Aldrich). Then, samples were washed with PBS containing 0.1% Triton X100 (three times for 15 min each time) at room temperature and incubated overnight at 4 °C with 14 µM 4′,6-diamidino-2-phenylindole (DAPI) in 1% BSA in PBS and the following secondary antibodies: Alexa-488 conjugated goat anti-rabbit IgG (1:500) (Jackson ImmunoResearch); Alexa-555 conjugated goat anti-mouse IgG (1:500) (A-21424, Thermo Fisher Scientific) and Phalloidin-iFluor 555 conjugate (1:200) (20552, Cayman Chemical). After washing with PBS containing 0.1% Triton X100 (three times for 15 min each time), the samples were mounted on slide glasses with SlowFade Gold antifade reagent (S36937, Thermo Fisher Scientific). For double staining of VWYa and NPWa, we used Zenon Rabbit IgG Labeling Kits (Z25300, Thermo Fisher Scientific) for direct labeling of anti-VWYa and anti-NPWa antibodies with Alexa-488 and Alexa-647, respectively, according to the manufacturer's instructions. After the blocking steps described above, 1 µg of each of the labelled antibodies was prepared and applied to the samples for 2 h at room temperature. After washing with PBST (three times for 15 min each time), samples were postfixed with 4% PFA in PBS at 4 °C overnight. Then, they were washed with PBS containing 0.1% Triton X100 (three times for 15 min each time) at room temperature, incubated with 1% BSA in PBS containing Phalloidin-iFluor 555 conjugate and 14 µM of DAPI for 2 h at room temperature. After washing with PBS containing 0.1% Triton X100 (three times for 15 min each time), they were mounted on slide glasses with Slowfade antifade reagent. The fluorescence of the samples was observed and images were recorded with a confocal microscopy system (SD-OSR, Olympus) and Metamorph v.7.10.1.161 (Molecular Devices). Images were edited with ImageJ (v.2.5.30/1.53f).

**Peptide functional assay.** A cydippid larva of *B. mikado* was transferred with seawater on to a glass slide with two spacers in parallel and covered with a coverslip. Seawater in the chamber was exchanged by draining seawater from one side with filter paper and adding fresh seawater from the other side. For NPWa functional analysis, the larva in the chamber was filmed under a microscope (BX53 and DP74, Olympus). The filming of *B. mikado* larvae was started 3 min before the start of the experiment. The medium was then replaced with seawater, seawater with the reverse peptide or seawater with the NPWa peptide with 3-min intervals, and finally filmed for 5 min after addition of seawater with the NPWa peptide. The concentration of the peptides ranged from 1 µM to 100 µM in tenfold increments. Adradial canal length was measured using ImageJ software. For each concentration (*n* = 5 for each), the average value from 10 frames (approximately 25 s) after each change of the medium was calculated and normalized by that obtained under seawater conditions. For VWYa functional analysis, a larva in the chamber was recorded 1 min before the start of the experiment, then 1.5 min after the change of medium to seawater, seawater with the reverse peptide or seawater with VWYa peptide. The concentrations of the peptides were 10 µM and 100 µM. The length of the oral half of the larvae was measured five times, and the average value was used for comparison. The shape of the oral half of the larvae was considered to be a hemisphere, and its volume was calculated and compared. The value was standardized by the average value obtained under seawater conditions.

**Neuropeptide–GPCR pair prediction.** Neuropeptide–GPCR pairs were predicted using the PD-incorporated SVM[37,38] prediction model, constructed with 2,467 compound–protein interactions (CPIs). The CPIs were collected from the IUPHAR database (https://doi.org/10.1111/bph.15538), UniProtKB (https://doi.org/10.1093/nar/gku989) and the literature (Supplementary Table 7) and consisted

of 932 human interactions, 384 mouse interactions, 568 vertebrate interactions and 581 invertebrate interactions. As the PD-incorporated SVM prediction model was constructed by training two-class SVMs[67], the non-interaction pairs were generated by randomly shuffling the GPCRs and peptides in the CPI data, and the peptide–GPCR pairs were represented as the linearized outer products of PDs[38] and transmembrane (TM) z scale descriptors[37]. For neuropeptide–GPCR prediction, AA sequences of GPCRs were acquired by scanning *M. leidyi* protein models with HMMER[52] (v.3.2.1) to detect GPCR-related Pfam motifs (7tm-1, 7tm-2 and 7tm-3) with an e value cut off of 0.01 (Supplementary Table 8). Of the collected protein models, GPCRs containing all seven TM domains were used for prediction. For neuropeptides, *M. leidyi* homologues of *B. mikado* neuropeptides were acquired by blastp search against *M. leidyi* protein models (Supplementary Data 1). Sixty-three peptides including length variants observed in *B. mikado* and 538 full-length GPCRs were converted to CPI descriptors and used as input to the constructed prediction model. The structures of *M. leidyi* neuropeptides and GPCR protein models used for the prediction can be found in Supplementary Table 4. The prediction model outputs the scores in a range from 0 (no interaction) to 1 (interaction). We chose a representative peptide for each peptide family using the following criteria. If no structural variation was observed among the peptides within the family, we selected the longest peptide to avoid including artificial short versions of peptides that could form during sample preparation. Otherwise, we performed sequence alignment and maximum-likelihood analysis in MEGA (v.7.0.26)[68] to visualize the positions of peptides in the tree. We then identified the peptide that displayed the most common structure in the family. There could be one or two representative peptides per family, depending on the number of peptides observed in the family. Based on the predicted peptide–receptor pairs, we selected putative neuropeptide GPCRs by the following steps. For each peptide, GPCRs were sorted from highest to lowest communication score. Then, we assigned a GPCR to a peptide if that GPCR had the highest score for that peptide. Otherwise, we unassigned that GPCR for that peptide. For the single-cell expression analysis, we considered at most five receptors per peptide. To determine putative neuropeptide cell targets, we combined by summation the normalized expressions of all GPCRs per peptide for each cell type.

**Reporting summary.** Further information on research design is available in the Nature Research Reporting Summary linked to this article.

## Data availability

Data generated or analysed during this study are included in the Supplementary Information. The mass spectrometry data and AA sequence data used for peptide identification have been deposited in the ProteomeXchange Consortium with data set identifier PXD030145. Other raw datasets associated with this study are available at https://doi.org/10.6084/m9.figshare.19930586.

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

## Acknowledgements

We thank the sequencing, scientific computing and data analysis, instrumental analysis and scientific imaging sections of the Okinawa Institute of Science and Technology Graduate University for their support with analyses. We thank N. Funayama for providing gemmules of *E. fluviatilis*. We also thank A. Dahal, A. Tanimoto, and J. Higuchi and M. Ohata for maintaining the *Nematostella* and ctenophore cultures. We thank S. Abe and M. Yorozu (Hakkeijima Sea Paradise), S. Ikeguchi, N. Hirata and M. Nagai (Notojima Aquarium), N. Ishikawa (Niigata City Aquarium Marinepia Nihonkai) and the staff of Misaki Marine Biological Station (University of Tokyo), the International Coastal Research Center, the Atmosphere and Ocean Research Institute (University of Tokyo), Oki Marine Biological Station (Shimane University), Shimoda Marine Research Center (University of Tsukuba) and Kiyoshi-Hiroshi (sea-grape farm) for their cooperation in the collection and rearing of ctenophores. This work was supported by JSPS KAKENHI (grant numbers JP20K06662 to H.W., JP19K06796 to E. H. and JP16H06280 to K.I.). C.G and O.H. were supported by a fellowship from JSPS DC1 (numbers JP19J20655 and JP19J20278, respectively).

## Author contributions

H.W. designed the research. O.H., C.K., E.K., K.J., K.S., K.I. and H.W. performed sample collection and culture. E.H. and C.K. performed mass spectrometry analysis. M.F.L. and S.S. performed transcriptome sequencing and assembly. M.F.L. performed the molecular phylogeny. C.G., O.H. and H.W. performed the transcriptome data analysis. Ryotaro Nakamura and Ryo Nakamura performed the gene knockdown. O.H., K.M., Ryotaro Nakamura, Ryo Nakamura and S.K. performed immunostaining and microscopic data analysis. A.S. and H.S. performed receptor prediction. O.H., K.S. and K.I. performed the neuropeptide assay. E.H., C.G., O.H. and H.W. aqcuired funding. E.H., C.G., O.H., A.S., K.M. and H.W. produced the manuscript draft and figures. All authors contributed to and approved the final version of the manuscript.

## Competing interests

The authors declare no competing interests.

## Additional information

**Extended data** is available for this paper at https://doi.org/10.1038/s41559-022-01835-7.

**Correspondence and requests for materials** should be addressed to Eisuke Hayakawa or Hiroshi Watanabe.

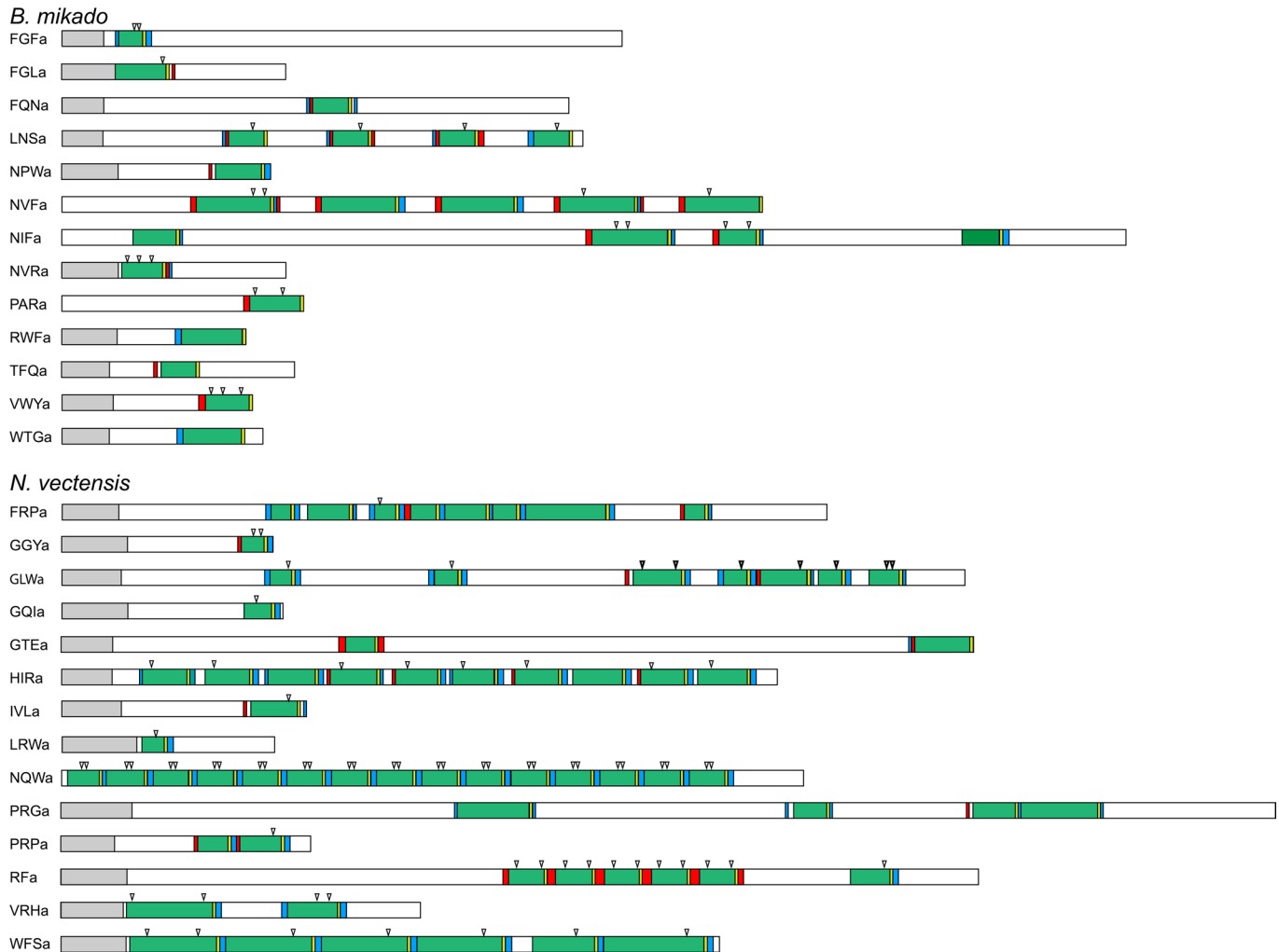

**Extended Data Fig. 1 | Structure overview of neuropeptide precursors identified.** Schematic representations of neuropeptide precursors identified in *B. mikado*. and *N. vectensis*. Grey boxes indicate the predicted signal peptide. Red, blue, yellow, and green boxes show acidic, basic cleavage sites, glycines as amide donor, and the regions encoding mature peptides, respectively. Triangles denote the putative cleavage sites of neprilysin endopeptidase.

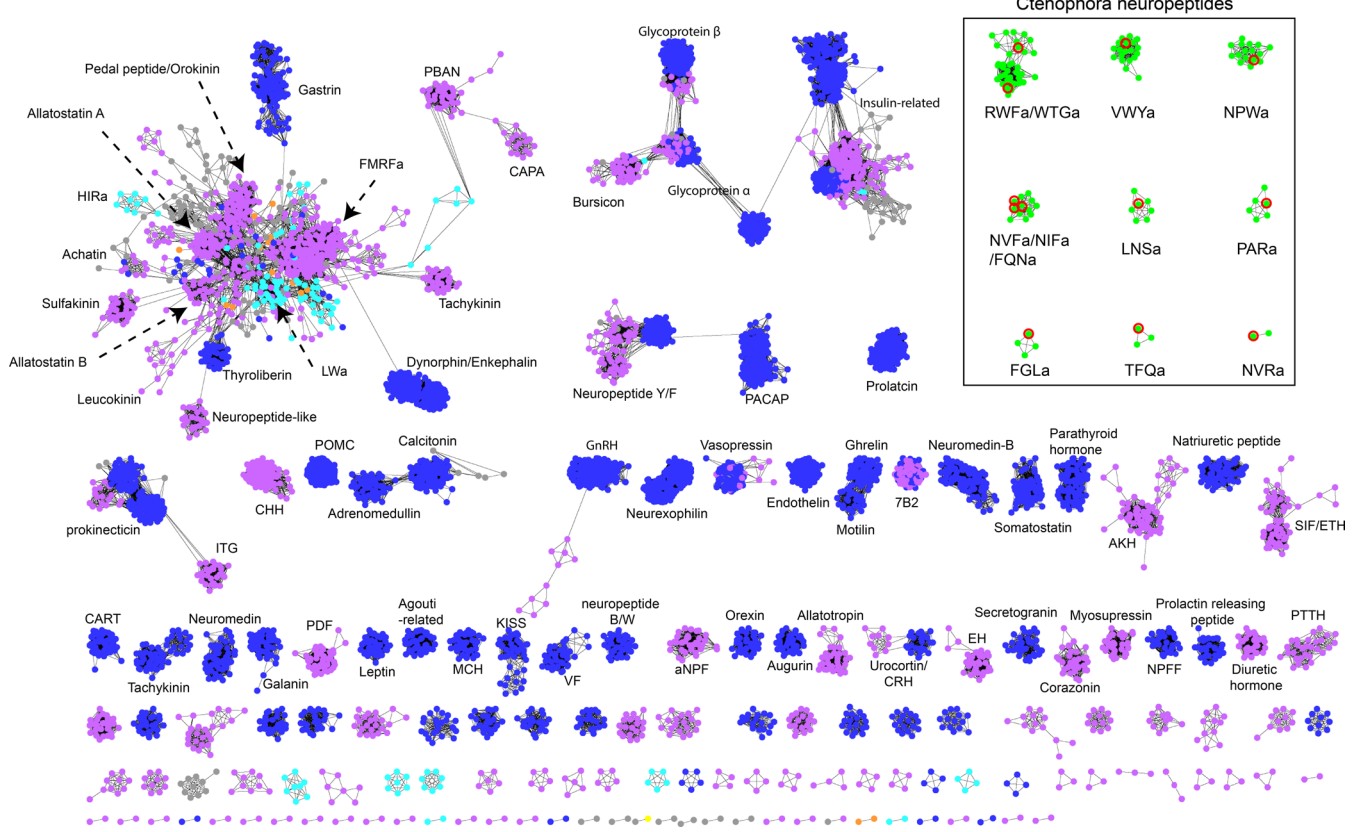

**Extended Data Fig. 2 | Global overview of neuropeptide precursor similarities in Metazoa.** Cluster map of neuropeptide precursors. Edges correspond to BLAST e-value of < 1e-5. Ctenophora: green, Cnidaria: light blue, Placozoa: orange, Xenoacoelomorpha: grey, Protostomia: purple, Deuterostomia: blue. *B. mikado* neuropeptides are indicated by red circles.

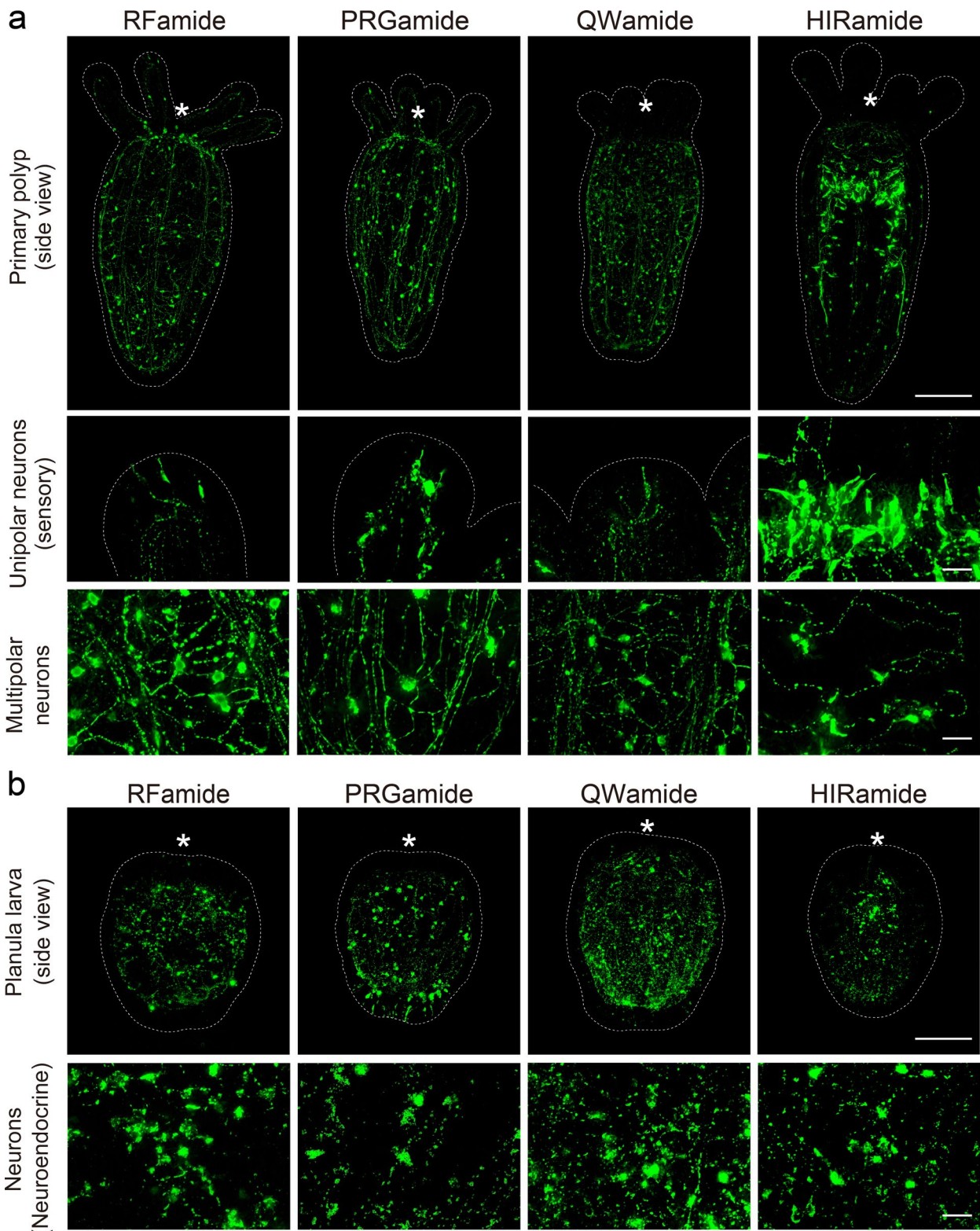

**Extended Data Fig. 3 | Immunostaining of neuropeptides in *N. vectensis*.** Expression patterns and subcellular localizations of RFa, PRGa, QWa, and HIRa of *N. vectensis*. **a**, Upper panels are side views of juvenile polyps stained by neuropeptide antibodies. Middle and lower panels show the neuronal morphologies and subcellular distributions of neuropeptides at higher magnification. Unipolar neurons with sensory cilia are located at the tentacle tip (RFa, PRGa, QWa) or the pharyngeal endomesoderm (HIRa). Multipolar neurons form subepithelial neural network (SNN) at the body column. **b**, Side views of late planula larvae (oral side: top). Neuropeptide-expressing neurons become visible from the larval stages. The neurons store neuropeptide-containing vesicle mainly at their cell bodies, or they do not have functional neurites, suggesting their neuroendocrine cell-like functions. Asterisks indicate oral positions. Large and small scale bars are 100 μm and 10 μm, respectively.

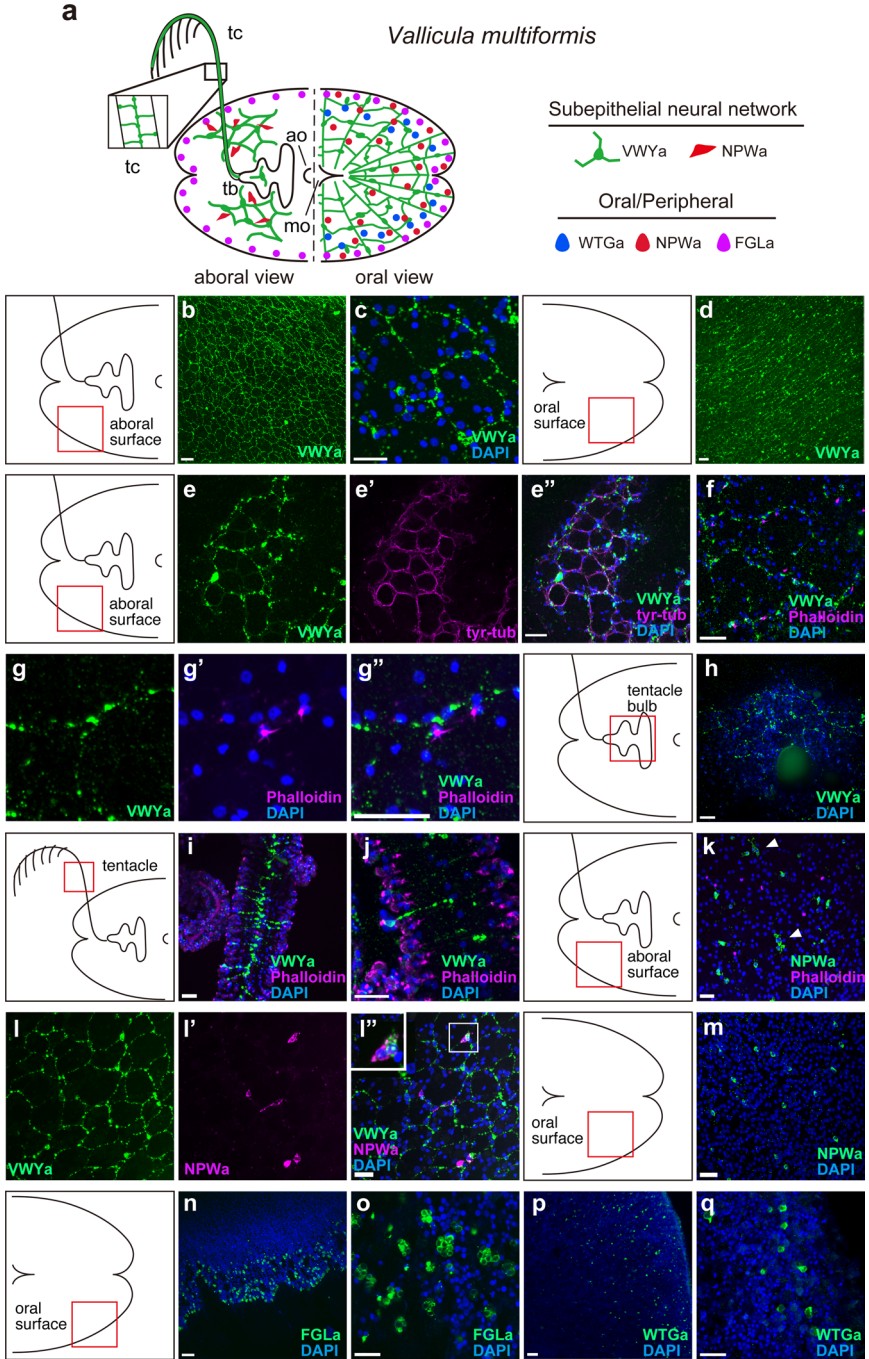

**Extended Data Fig. 4 | Immunostaining of peptides in *V. multiformis*.** Neuropeptide immunofluorescent staining of adult *V. multiformis*. **a**, Results of immunofluorescent are summarized as in the schematic diagram of *V. multiformis*. The left half shows the aboral view and the right half shows the oral. Abbreviations: ao, apical organ; tc, tentacle; tb, tentacle bulb; mo, mouth. The distributions of each peptide-expressing cell type are indicated in different colors. **b–d**, subepithelial neural network on the aboral surface (**b**, magnified view in **c**) and the oral surface (**d**) are visualised by the VWYa staining. **e–e″**, VWYa+ neurons (green) develop polygonal neural mesh in the dorsal subepithelial layer where tubulin staining (magenta) is partially overlapped. **f**, Distribution of VWYa positive neural net-like structures (green) and phalloidin positive protrusions (magenta) on the aboral surface. **g–g″**, Some of these protrusions (magenta) are extended from the VWYa+ cell bodies (green). **h**, This neural network-like pattern was observed on the surface of the tentacle bulb. **i**, VWYa+ neural network-like pattern (green) was also observed in the tentacle. **j**, Cell bodies of VWYa+ neurons (green) bearing F-actin (phalloidin)-positive protrusion (magenta) are located on the surface of the tentacle. **k,m**, NPWa+ cells distributed sparsely on the aboral (**k**) and oral (**m**) epithelial surface. The distribution of NPWa was not observed on the neurite-like structure, but in the short processes in some of the positive cells, especially on the aboral side (arrowheads in **k**). NPWa+ cells (green in **k**) are not associated with phalloidin positive protrusion of F-actin (magenta in **k**). **l–l″**, Double immunostaining of VWYa and NPWa. The distribution of VWYa+ (green) and NPWa+ (magenta) were not overlapped but cells positive for each peptide were closely associated (magnified in inset). **n,o**, Distribution of FGLa+ cells was observed on the body margin. FGLa+ cells contained large spherical organelles in their cell bodies (**o**). **p,q**, Distribution of WTGa. WTGa+ cells were small round shapes and sparsely distributed on the oral surface of the animal. The frequency of these cells was higher in the outer region. Schematic diagrams inserted in the panels indicate the position of the following images in the animal. Scale bars are 20 μm (**c**, **e–o** and **q**) and 50 μm (**b**, **d** and **p**). Pseudo-colours in **b–q** were applied by ImageJ software.

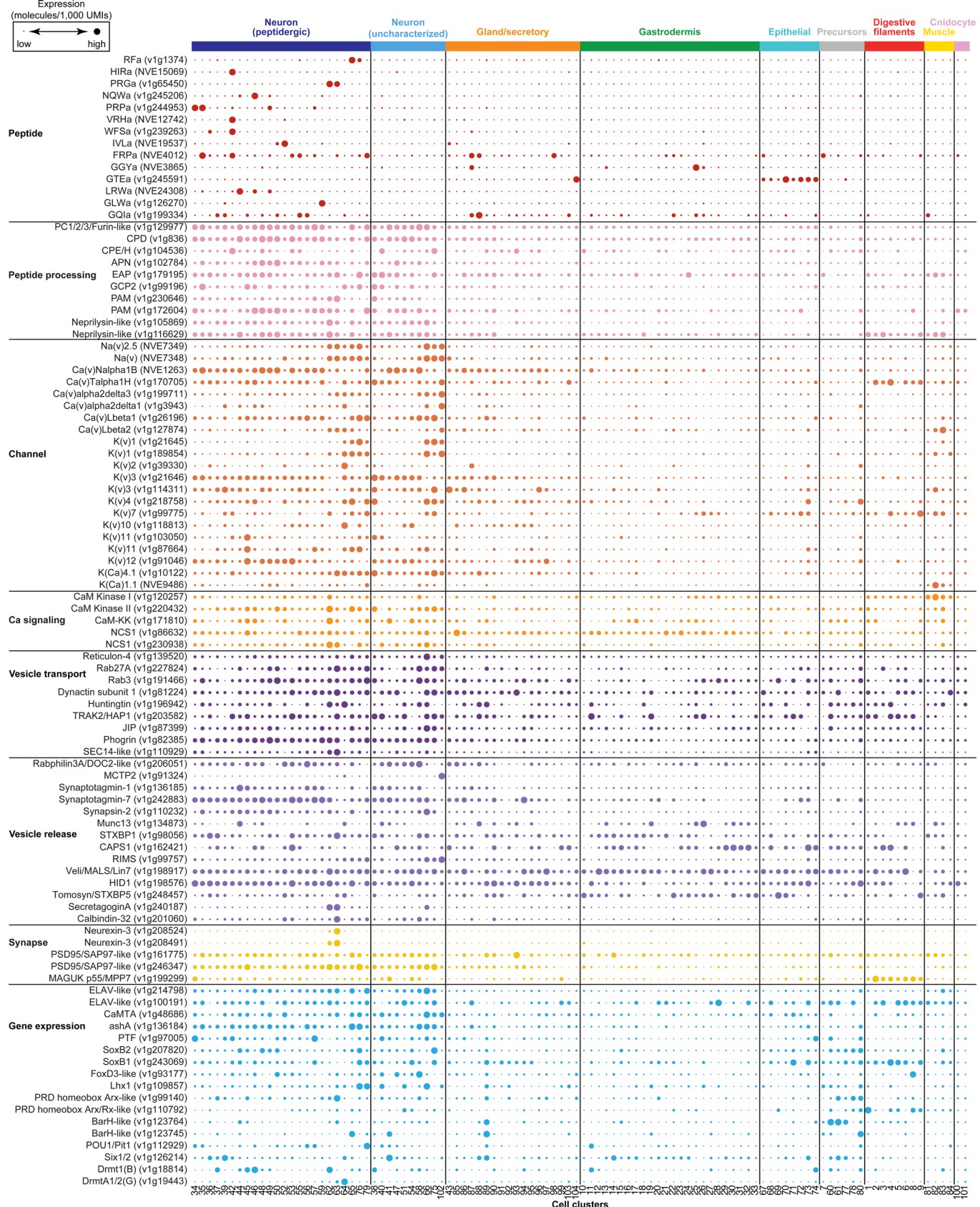

**Extended Data Fig. 5 | Genetic signature of peptidergic cells in adult *N. vectensis*.** Expression of gene homologs involved in peptidergic neuronal function across the different cell clusters of adult *N. vectensis* scRNA-seq data[27]. The normalized expression values (molecules/1,000 UMIs) are scaled by gene with dot size scales from smallest to largest, corresponding to lowest and highest expression, respectively. Dots are colored according to gene functional categories.

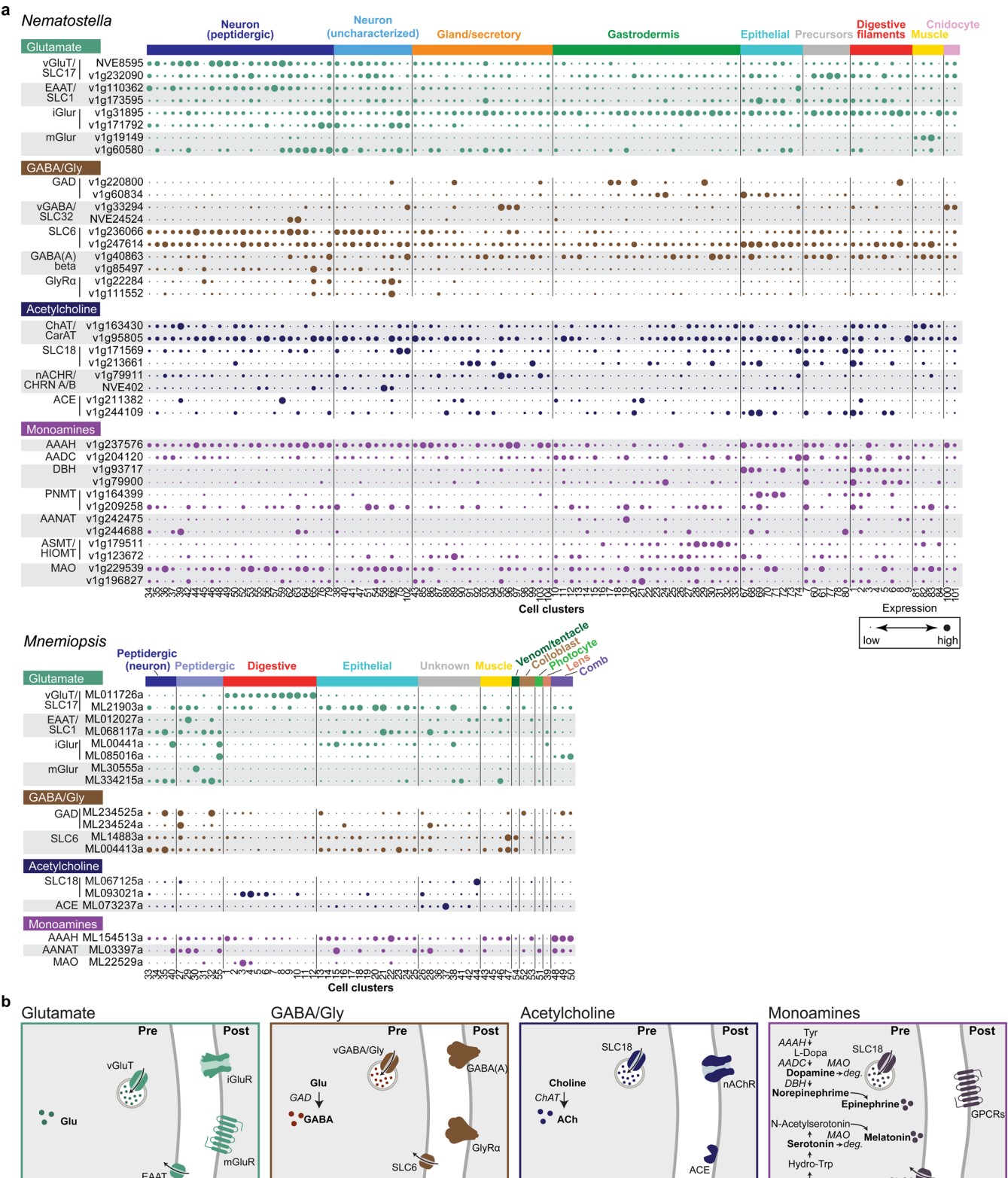

**Extended Data Fig. 6 | Gene expression profile of chemical neurotransmission machineries in *N. vectensis* and *M. leidyi*. a**, Expression of gene homologs involved in neurotransmitter production, secretion, and reception across the different cell types of adult *N. vectensis* and *M. leidyi* scRNA-seq data[12,27]. The normalized expression values (molecules/1,000 UMIs for *N. vectensis*; molecules/10,000 UMIs for *M. leidyi*) are scaled by gene with dot size scales from smallest to largest, corresponding to lowest and highest expression, respectively. **b**, Schematic representations of the different neurotransmitter signaling pathways (glutamate, GABA/glycine, acetylcholine and monoamine).

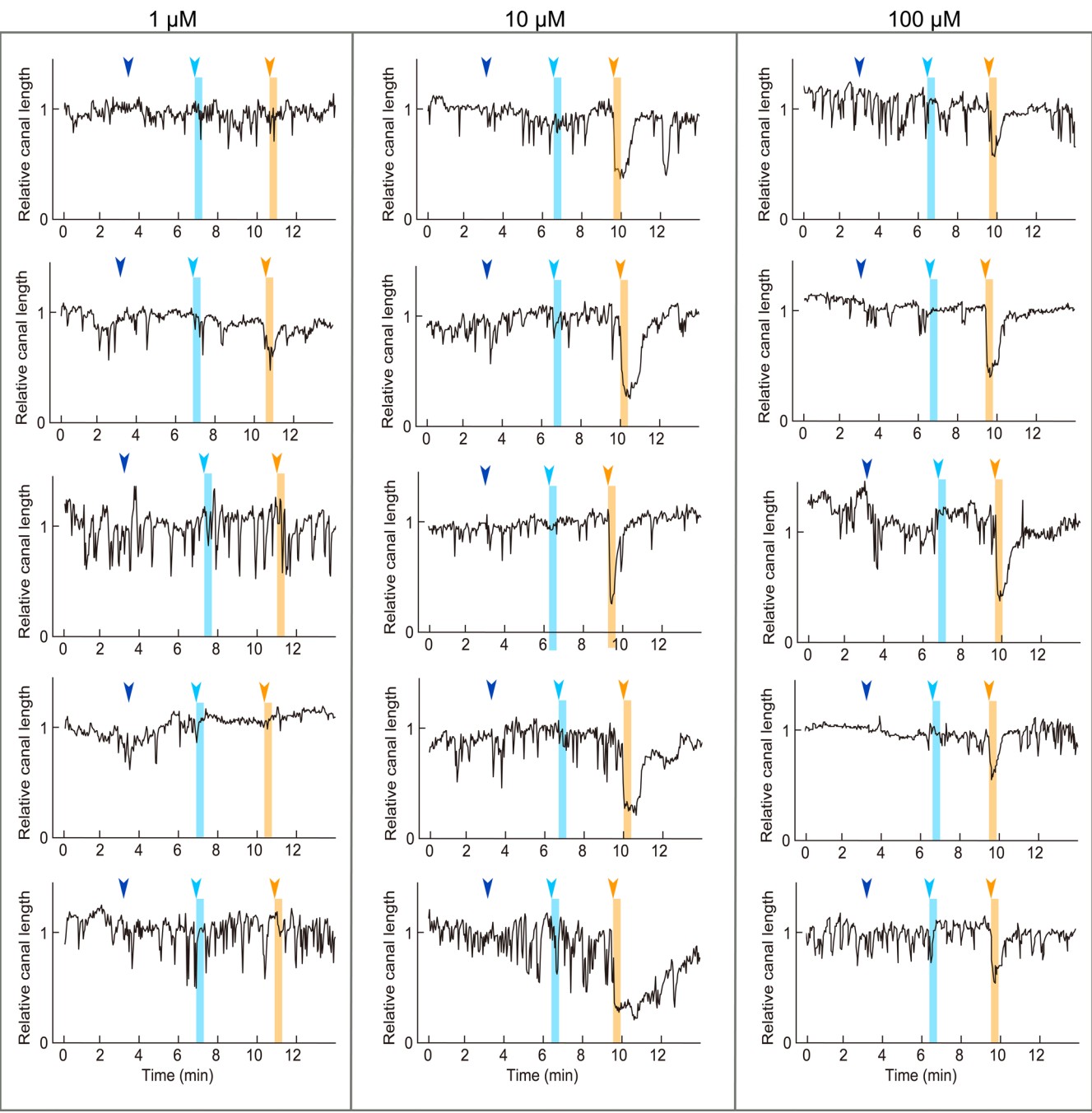

**Extended Data Fig. 7 | Effect of NPWa peptide treatment on adradial canal contraction of _B. mikado_.** Full time series of _B. mikado_ reaction to increasing concentrations of NPWa peptide. The line graphs show the relative length of the adradial canal standardized by the mean of the seawater conditions. The arrowheads indicate the time point at which the seawater medium was changed to fresh seawater, reverse peptide-, or NPWa peptide-containing seawater (blue, light blue, and orange, respectively). The concentrations of peptides are shown at the top. The mean values of the time period shaded in light blue and orange were calculated for the quantitative analysis (Fig. 4c).

# Reporting Summary

## Statistics

For all statistical analyses, confirm that the following items are present in the figure legend, table legend, main text, or Methods section.

| n/a | Confirmed | |
|---|---|---|
| ☐ | ☒ | The exact sample size (*n*) for each experimental group/condition, given as a discrete number and unit of measurement |
| ☐ | ☒ | A statement on whether measurements were taken from distinct samples or whether the same sample was measured repeatedly |
| ☐ | ☒ | The statistical test(s) used AND whether they are one- or two-sided *Only common tests should be described solely by name; describe more complex techniques in the Methods section.* |
| ☒ | ☐ | A description of all covariates tested |
| ☐ | ☒ | A description of any assumptions or corrections, such as tests of normality and adjustment for multiple comparisons |
| ☐ | ☒ | A full description of the statistical parameters including central tendency (e.g. means) or other basic estimates (e.g. regression coefficient) AND variation (e.g. standard deviation) or associated estimates of uncertainty (e.g. confidence intervals) |
| ☐ | ☒ | For null hypothesis testing, the test statistic (e.g. *F*, *t*, *r*) with confidence intervals, effect sizes, degrees of freedom and *P* value noted *Give P values as exact values whenever suitable.* |
| ☒ | ☐ | For Bayesian analysis, information on the choice of priors and Markov chain Monte Carlo settings |
| ☒ | ☐ | For hierarchical and complex designs, identification of the appropriate level for tests and full reporting of outcomes |
| ☒ | ☐ | Estimates of effect sizes (e.g. Cohen's *d*, Pearson's *r*), indicating how they were calculated |

*Our web collection on statistics for biologists contains articles on many of the points above.*

## Software and code

Policy information about availability of computer code

Data collection    Fusion Lumos (Thermo Fisher Scientific) with Xcalibur (version 4.5, Thermo Fisher Scientific) was used for mass spectrometry. RNA-seq was performed by using HiSeq 2500 (Illumina). Fluorescent microscope images of N. vectensis was acquired using ECLIPSE Ni (Nikon) equipped with Ds-Ri2 (Nikon). Immunostaining images of B. Mikado were acquired using confocal microscopy (Olympus Fluoview FV10i). Immunostaining images of V. multiformis were recorded with a confocal microscopy system SD-OSR (Olympus) handled by the software Metamorph (Molecular devices, ver. 7.10.1.161).

Data analysis    ProteoWizard (package 3.0.20139) was used to convert mass spectrometry data. Peptide-to-spectrum matching was performed using PEAKS X software (PEAKS Studio version 10.0, Bioinformatics Solutions) and Mascot (version 2.7, Matrix Science). Protein motif scan was performed using HMMER (version 3.2.1). Protein homology search was performed using BLAST blastp, version 2.11.0 +). Network visualization of neuropeptide precursor proteins were done using Cytoscape (version 3.5.1), The transcriptome data was processed using libngs (https://github.com/sylvainforet/libngs), Trinity (v2.3.2), CD-HIT (v4.6.5) and TransDecoder (v3.0.1). Sequence alignment was performed using MEGA (version 7.0.26). Codes used for gene expression profiling analysis: https://github.com/oist/scrna-counts. Fluorescent microscope images of N. vectensis was processed with NIS-Elements AR (Version 5.02.03, Nikon). Image data were edited using ImageJ (ver. 2.5.30/1.53f).

For manuscripts utilizing custom algorithms or software that are central to the research but not yet described in published literature, software must be made available to editors and reviewers. We strongly encourage code deposition in a community repository (e.g. GitHub). See the Nature Portfolio guidelines for submitting code & software for further information.

## Data

Policy information about availability of data

All manuscripts must include a data availability statement. This statement should provide the following information, where applicable:
- Accession codes, unique identifiers, or web links for publicly available datasets
- A description of any restrictions on data availability
- For clinical datasets or third party data, please ensure that the statement adheres to our policy

> The mass spectrometry data ,AA sequence data used for peptide identification and other resources related to peptide identification have been deposited to the ProteomeXchange Consortium with the dataset identifier PXD030145 (https://repository.jpostdb.org/preview/194169514461ab01a6d09da, Access key : 1700).

# Field-specific reporting

Please select the one below that is the best fit for your research. If you are not sure, read the appropriate sections before making your selection.

☒ Life sciences ☐ Behavioural & social sciences ☐ Ecological, evolutionary & environmental sciences

For a reference copy of the document with all sections, see nature.com/documents/nr-reporting-summary-flat.pdf

# Life sciences study design

All studies must disclose on these points even when the disclosure is negative.

| | |
|---|---|
| Sample size | For mass spectrometry analysis, we prepared four samples independently for each animal species. For peptide biological assays, we examined five biologically independent samples. |
| Data exclusions | No data were excluded. |
| Replication | For mass spectrometry, three samples for each animal species were directly subjected to LC-MS/MS analysis and one sample was further fractionated to reduce complexity. For peptide biological assays, we examined five biologically independent samples. |
| Randomization | For peptide biological assays, we picked up Cydippid larvae randomly from same batch of the larvae. |
| Blinding | All measurements in peptide biological assays were collected blind to the identify of peptide sequences and concentrations. |

# Reporting for specific materials, systems and methods

We require information from authors about some types of materials, experimental systems and methods used in many studies. Here, indicate whether each material, system or method listed is relevant to your study. If you are not sure if a list item applies to your research, read the appropriate section before selecting a response.

## Materials & experimental systems

| n/a | Involved in the study |
|---|---|
| ☐ | ☒ Antibodies |
| ☒ | ☐ Eukaryotic cell lines |
| ☒ | ☐ Palaeontology and archaeology |
| ☐ | ☒ Animals and other organisms |
| ☒ | ☐ Human research participants |
| ☒ | ☐ Clinical data |
| ☒ | ☐ Dual use research of concern |

## Methods

| n/a | Involved in the study |
|---|---|
| ☒ | ☐ ChIP-seq |
| ☒ | ☐ Flow cytometry |
| ☒ | ☐ MRI-based neuroimaging |

## Antibodies

| | |
|---|---|
| Antibodies used | All neuropeptide antibodies were generated by ourself.<br>The following commercial antibodies were used : Alexa488-conjugated anti-rabbit secondary antibody (1:500) (111-545-003, Jackson Immuno Research LABORATORIES), Alexa-488 conjugated goat anti-rabbit IgG (1:500) (Jackson ImmunoResearch), Alexa488 conjugated anti-rabbit secondary antibody (1:500) (A-11008, Thermo Fisher Scientific), anti-tyrosinated tubulin, (1:500) (T9028, Sigma-Aldrich). |
| Validation | Manufacturer validation statements can be found below for the corresponding antibodies.<br>Alexa488-conjugated anti-rabbit secondary antibody (111-545-003, Jackson Immuno Research LABORATORIES) : https://www.jacksonimmuno.com/catalog/products/111-545-003<br>Alexa488 conjugated anti-rabbit secondary antibody (A-11008, Thermo Fisher Scientific) : https://www.thermofisher.com/antibody/ |

product/Goat-anti-Rabbit-IgG-H-L-Cross-Adsorbed-Secondary-Antibody-Polyclonal/A-11008
anti-tyrosinated tubulin (T9028, Sigma-Aldrich) : https://www.sigmaaldrich.com/product/sigma/t9028

# Animals and other organisms

Policy information about studies involving animals; ARRIVE guidelines recommended for reporting animal research

| | |
|---|---|
| Laboratory animals | laboratory culture of animals (Nematostella vectensis, Bolinopsis mikado, Vallicula multiformis, and Ephydatia fluviatilis were maintained at the Okinawa Institute of Science and Technology Graduate University. Bolinopsis mikado were cultured also at the Shimoda Marine Research Center, University of Tsukuba. |
| Wild animals | Bolinopsis mikado were collected at Kamo Bay (Oki island, Shimane), Hakkeijima (Yokohama, Kanagawa), Nanao Bay (Nanao, Ishikawa), and Tabira Bay (Hirado, Nagasaki) in Japan. Bolinopsis collections were done by gently surrounding a sample with a long (2-5 m) griped plastic cap. Neuroepetides were extracted from adult Bolinopsis. For neuropeptide assays, we didn't use wild animals. Vallicula multiformis were collected at a sea-grapes farm Kiyoshi-Hiroshi at Ginoza (Okinawa, Japan). |
| Field-collected samples | Bolinopsis mikado were maintained at 20 °C in a 60 l aquarium with slow water circulation. They were fed artemia in the morning and evening, with two to three feedings of frozen copepod (Pacific Trading and Kyorin) in between. Vallicula multiformis were maintained in the 1.5 l seawater at 25 °C with feeding freshly hatched artemia two times per week. |
| Ethics oversight | No ethics oversight was required due to the nature of the organisms. |

Note that full information on the approval of the study protocol must also be provided in the manuscript.

