## [Peer Review File · Nature Ecology & Evolution]

Peer Review Information

Journal: Nature Ecology and Evolution

Manuscript Title: Mass spectrometry of short peptides reveals common features of metazoan peptidergic neurons

Corresponding author name(s): Hiroshi Watanabe

Editorial Notes:

Reviewer Comments & Decisions:

Decision Letter, initial version:
--

Dear Professor Watanabe,

Your manuscript entitled "Common Features of Metazoan Peptidergic Neurons Support Their Single Origin" has now been seen by three reviewers, whose comments are attached. The reviewers have raised a number of concerns which will need to be addressed before we can offer publication in Nature Ecology & Evolution. We will therefore need to see your responses to the criticisms raised and to some editorial concerns, along with a revised manuscript, before we can reach a final decision regarding publication.

We therefore invite you to revise your manuscript taking into account all reviewer and editor comments. Please highlight all changes in the manuscript text file in Microsoft Word format.

* If you have not done so already please begin to revise your manuscript so that it conforms to our Article format instructions at <http://www.nature.com/natecolevol/info/final-submission>. Refer also to any guidelines provided in this letter.

{redacted}

Nature Ecology & Evolution is committed to improving transparency in authorship. As part of our efforts in this direction, we are now requesting that all authors identified as 'corresponding author' on published papers create and link their Open Researcher and Contributor Identifier (ORCID) with their account on the Manuscript Tracking System (MTS), prior to acceptance. ORCID helps the scientific community achieve unambiguous attribution of all scholarly contributions. You can create and link your ORCID from the home page of the MTS by clicking on 'Modify my Springer Nature account'. For more information please visit www.springernature.com/orcid.

Yours sincerely,

{redacted}

Reviewer expertise:

Reviewer #1: evo-devo of the cnidarian nervous system

Reviewer #2: cnidarian and ctenophore nervous system

Reviewer #3: neurochemistry, mass-spectrometry

Reviewers' comments:

2Reviewer #1 (Remarks to the Author):

The nervous system of ctenophores has recently attracted a lot of attention due to the seemingly low degree of conservation of its molecular constituents compared to the nervous systems of cnidarians and bilaterians. These observations prompted speculations that the ctenophore nervous system may have evolved “independently”, though it is often not clear what is meant by this term. In this study, the authors address a specific aspect of this fascinating topic, the molecular make-up of peptidergic neurons in a ctenophore (*B.mikado*), a cnidarian (*N.vectensis*) and a sponge (*E. fluviatilis*). Neuropeptides are identified by mass spec and analysis of their sequences shows that the *B.mikado* neuropeptides are not conserved in other phyla, and not or poorly conserved in other ctenophores (in which they have mainly been predicted from genome sequences). Using antibodies against six neuropeptides, the authors document the general distribution of cells expressing these NPs, some of them in cells with long processes typical for neurons. The manuscript then switches to an analysis of existing single cell RNAseq data to show that cells expressing neuropeptides also express genes involved in the processing, transport and release of vesicles. Treatment with two of the peptides is shown to affect the contraction of different structures in *B.mikado*, the more aborally located adradial canals and the oral epithelium, respectively. Finally, the authors use computational predictions to suggest potential ligand-receptor pairs and then analyse the expression of the putative receptors based on existing scRNAseq data from *Mnemiopsis*.

The main strengths of the manuscript are the biochemical identification of neuropeptides (as compared to computational prediction), the use of newly generated antibodies to describe populations of neurons in *B. Mikado* that indeed express some of the neuropeptides, and the observation of effects upon treatment with synthetic neuropeptides. The manuscript further explores the published single cell RNAseq data for *Mnemiopsis* and *Nematostella* to support the neural identity of cell expressing putative neuropeptides. Though the use of *Bolinopsis* for peptide identification and *Mnemiopsis* for analysis of single cell data makes some comparisons less convincing, I find it positive that a less well characterized ctenophore species is added here. Overall, the paper provides further support for the notion that peptidergic neurons were present at early stages of nervous system evolution, as has most recently be suggested by Sachkova et al., 2021. The data and analyses presented here are certainly of broad interest for the ongoing debate about the evolutionary origin(s) of neurons and nervous systems and help moving this debate from a general and rather superficial level to more specific features of neural physiology.

There are a several points that need to be addressed in a revised version.

1. “Neuropeptides” are defined by structural criteria, both in computational and here in biochemical approaches. Their expression in neurons has to be demonstrated for each putative “neuro”peptide and the presence of a “neuro”peptide does not identify a cell as a neuron, in particular not in animals that are an outgroup to other animals with neurons. My impression is that this is not consequently observed here. In Fig 2a and b, NPWa and VWYa are shown in cells with a typical neural morphology. In Fig2c, FGLa, WTGa, RWFa and GVFa-positive cells are described as sensory and endocrine-like neurons. But beyond the expression of these peptides, what is the basis for assuming that these are neural cells? This classification then appears to be used to label cells in *Mnemiopsis* (a different

3ctenophore) as “Neuron (peptidergic)” in Ext Fig 7 and to collapse them into one group “neuron(peptidergic)” in Fig 3. The presence of the processing and secretion machinery does not provide independent support for this assumption, as I would think that this machinery needs to be present in any cell that expresses these peptides, whether they are neural or not. “Synapse” and “gene expression” genes appear to be similar to other cells, in particular “epithelial”. The criteria for calling a cell neural need to be clearly stated in the main text. If it is the expression of a “neuro”peptide, the issue could be addressed by separating those cells/metacells for which the antibody stainings (here or in Sachkova et al.) have shown neural morphology from those for which only the presence of the “neuro”peptide is used for their classification. They could be labelled potential neural cells or just peptidergic cells. My apologies in case I didn’t follow the authors rationale correctly, but if so, it would be good to explain it in more detail.

In the same vein: why are Mnemiopsis cl 27 and 29 listed as “Neuron (peptidergic)” in the same Ext Fig 7? Epithelial cluster 18 and 19 seem to be very similar.

2. From line 205 on, the expression of transcription factors is used to show similarities to the regulatory program of bilaterian neurons. I didn’t understand how the TFs in this list were selected. Would there be paralogs (other bHLH or soxB genes) that are not in the “neuron(peptidergic)” clusters but enriched in other cell types? Since some of these TFs (e.g. sox1, BarHx, six1/2) are rather broadly expressed, I was wondering about the strength of this correlation and whether it would be possible to use different TFs to make a different point (e.g. the peptide expressing cells are muscle cells or epithelial cells).

3. for Fig 2d, please describe better or use a cartoon to show where in the animals the imaged area is.

4. In line 163, stereocilia are mentioned. I cannot discern stereocilia or any other type of cell protrusion in Fig 2d. This is relevant for describing these cells as sensory in the figure legend. Please provide a better image or change the text to remove stereocilia/cilia. Please change line 437 (figure legend to putative sensory cell, even if you can demonstrate stereocilia.

5. Line 175: a “close” species: please provide a reference for ctenophore phylogeny. “Close” is very subjective, also given the moderate level of overlap in the neuropeptide repertoire.

6. Line 194: The labelling indeed seems to be not restricted to typical synaptic puncta. But I don’t think there is homogenous distribution of the neuropeptide signal. “Broad” would be better.

7. Line 201-202: This seems to assume that synapses should have the same molecular composition as those in bilaterians, which is one of the contentious points for the question about the similarity of the ctenophore neurons.

8. Whenever possible (e.g.. line 199) please avoid using “ctenophore” and indicate the species.

9. Line 231: The lack of labelling in Fig2 makes it a bit difficult to relate the expression of NPW to the observed effect of the peptide treatment. Is it possible to indicate the position of the mouth in Fig 2?

10. Maybe I missed it and it is in the supplement: A table for comparing the Mnemiopsis and

4Bolinopsis neuropeptides would be nice, even though the methods for their identification differ.

11. At what age/stage of Bolinopsis was mass spec performed?
12. Are there any controls for the antibodies? Like pre-adsorption with antigen? Or Western?
13. What is the counterstain in Ext Fig 3?
14. Do the acidic cleavage sites require different proteases? If so, is this included in the analysis of the Mnemiopsis clusters?
15. The more philosophical question: do the data really provide much support for a common origin of neurons? The peptides appear to have very little conservation compared to other animals (and even among ctenophores), so I am wondering if we would consider it support for common origin if ctenophores would use a monoamine that is not used in any other nervous system. No need to answer this, but I would argue that the data are interesting irrespective of their implications for answering a question that might be rooted mainly in semantics.

Reviewer #2 (Remarks to the Author):

The manuscript by Hayakawa E. and co-authors addresses a very important and currently highly debated topic – the evolutionary origin of the nervous system. This manuscript reports new data which will definitely attract significant attention of the community interested in evolution of neurons and neuropeptides. To characterise neurons in two non-bilaterian lineages, Ctenophora and Cnidaria, authors performed a comprehensive survey of neuropeptides in *Bolinopsis mikado* and *Nematostella vectensis*. Importantly, to identify neuropeptides authors used a completely unbiased approach – mass spectrometry. Unlike earlier studies of ctenophore neuropeptide repertoire that predicted neuropeptide precursors based on features known from bilaterian animals, this manuscript reveals unique mature structures of ctenophore-specific neuropeptides for the first time. Moreover, the authors reveal that the maturation process in non-bilaterians, especially in Ctenophora, occurs differently from bilaterian animals. Physiological role was also demonstrated for two of the new neuropeptides in an elegant functional test. Authors also revealed that newly discovered neuropeptides are produced by nerve nets and sensory cell in both ctenophora and cnidaria. I have however several concerns and suggestions (please see below). I think after addressing my comments the paper should be accepted for publication.

Comments:

1. The molecular identity of ctenophore neurons has been reported recently by Sachkova et al, 2021. Specifically, C33 cell cluster (from Sebe-Pedros et al, 2018) was identified as a neuron of the subepithelial neural net (SNN), C34 – neuron of the pharyngeal neural net, C35 - sensory neurons located in the gut and around the mouth. Additionally, C27, C30, C31, C32, and C40 cell clusters were identified as sensory cells of the aboral organ and C55 as a peptidergic tentacle cell. Thus, transcriptomes of several peptidergic neurons and sensory cells have been revealed already (please see Data S2 and Figure 5a in Sachkova et al, 2021). I think this needs to be discussed in the

5manuscript.

2. Localisation of ctenophore neuropeptides by immunostaining worked really nicely. I think it would be worth to discuss conservation of neuropeptide expression patterns between *B. mikado*, *V. multiformis* and *Mnemiopsis*. WTGa corresponds to ML02212a from *Mnemiopsis*, its localisation to SNN and aboral organ has been shown by both ISH and immunostaining by Sachkova et al, 2021 (please see Figure 2). Additionally, localisation of VWYa (ML02736a) to SNN and FGLamide (ML30511a) to aboral organ have been revealed by ISH (Sachkova et al, 2021, Figure 2, Fig S5). It would be also interesting to discuss the conservation of neuronal architecture: peptidergic SNN and neurons beneath the comb rows and ciliated grooves were reported in *Mnemiopsis* as well.

However I have a concern regarding the specificity of antibodies used for *Nematostella* and NPWa antibody. Peptides used for immunisation are just 2 to 3 residues long, it may result in non-specific binding to several targets. This choice of short peptides needs to be explained.

3. Analysis of peptidergic cells' transcriptome (Fig 3): "The dot plots show the summed normalized expression of the highest-expression gene (Methods) from all the cell clusters of each cell category (bottom)." The rationale behind this approach should be explained better since the logic of summing up gene expression levels across several clusters is not clear. Because neuropeptide-positive clusters might have significantly different gene expression profiles (e.g., one gene may be upregulated in one cell type but not detectable in another one), this approach may create artificial hybrid cell types. For example, RIMS (ML017713a) and Munc13 (ML24335a) are enriched in C35 but were not detected in C40. Another example is Secretagoin (ML03617a) enriched in C33 but not detected in C29 and C30. This approach may also create an additional bias due to different number of cell clusters included into each category. For example, Elav (ML220720a), is highly expressed in comb metacells (C49-C50) however after summing up across the categories it appears that total expression is higher in Neurons (10 clusters) and not in Comb (3 clusters).

5. "Digestive neuron" is an interesting hypothesis. I suggest to discuss it in comparison to other recently proposed hypotheses. For example, Moroz et al, 2021 proposed that neurons evolved multiple times from unrelated types of secretory cells. Further, the "divide-and-conquer model" by Burkhardt & Jekely, 2021 suggested that ctenophores have several neuronal cell types that probably evolved from the diverse secretory cells, and only one of them (sensory neurons around the ctenophore mouth, C35) is homologous to bilaterian neurons.

6. Synapses with translucent synaptic vesicles do exist in ctenophores (Hernandez-Nicaise 1973); translucent vesicles are normally filled with small molecule transmitters. The same neuron of the nerve net has both peptidergic and synaptic vesicles (Sachkova et al, 2021). I think these facts need to be considered when discussing pre- and postsynaptic machinery (Line 202: «the modest equipment of pre- and postsynaptic machinery, suggests that the peptidergic signaling in Ctenophora is mediated by volume transmission») and the role of glutamate ("Given that glutamate is secreted mainly from the non-neural cells in Ctenophora, the network-dependent functionality of the nervous system might be an evolutionarily secondary system").

7. Line 151: "... if they exist, the concentration should be lower than the other basal metazoans, which is consistent with the reduced complexity of the peptide receptor repertoire in Porifera..." – this statement is not really clear. I think the reduced complexity of peptide receptor repertoire rather correlates with reduced number of peptides, not their concentration. Additionally, I would suggest to rephrase "peptide receptor" as "peptide receptor homologues" since it has not been shown experimentally that they indeed bind peptides.

8. Line 543 – 544: "The complete list of identified peptides is provided in Supplementary Table 1 and

62." In addition to the list of peptides of interest provided in these tables, full list of proteins identified by mass spectrometry should be made available. An easily accessible .csv file with full search engine output for all the studied organisms (including the sponge) is critical for reproducibility of this research.

9. Line 687 – 688: "If structural variation is not observed among the peptides within the family, we selected the longest." It would be nice to explain the rationale behind choosing the longest variant. It is possible that after precursor cleavage at specific sites, peptides undergo further cleavage by aminopeptidases and therefore the longest variant would represent an intermediate step in the maturation rather than a final product.

10. Line 397: «small molecule fractions» - I guess authors meant peptide fractions.

11. Recent preprint by Yañez-Guerra et al, 2021 reported 2 neuropeptide precursors (phoenexin and nesphatin) conserved in sponges. I think it's worth discussing in connection to the fact that mass spectrometry did not identify these peptides.

Reviewer #3 (Remarks to the Author):

This is an impressive and well performed study. The LC-MS peptidomics measurements are very well performed and described, and the author appreciates the deposition of the data to make it public access. I enjoyed reading the article. A few questions and points below.

(1) As a minor point, they report that in addition to the well-known basic residue cleavages in prohormones, they observe acidic (D/E) residues are cleaved in many of their precursors. While this is likely correct, there have been past reports that under acidic storage conditions, aspartic acid and some other residues in peptides are cleaved, and fraction cleaved depending on the surrounding amino acid sequences. Could their observation of acidic cleavages, at least partially, be a sample preparation artifact as they stored their peptides in a formic acid solution?

(2) While amidation is certainly a hallmark of bioactive neuropeptides (and toxins: see below), in most animals, only a fraction of bioactive neuropeptides are amidated; their search algorithm appears to require this. Any thoughts on what fraction of peptides this excluded? Similarly, did they find a recognizable PAM enzyme in all the organisms studied? Or was it the related PHM and PHL duo which combine to have the same function?

(3) Many "neurotransmitter" predate neurons but are still cell secreted factors; bacteria use dopamine and other catecholamines as quorum sensing molecules, both bacteria and plants use serotonin (it is a known auxin), they are growth factors in animals, etc. The statements made throughout assume neurotransmitters are from neurons. Perhaps these statements should be modified slightly in terms of their evolutionary wide-spread non-neuronal functions.

(4) The ancient roles of some of these molecules become important when thinking about prohormone products as neuropeptides / hormones. Yeast process and secrete peptides for mating. In animals, there are many alternative uses for peptides from prohormones. One is to create peptide toxins, with some reports claiming this was the initial function of some prohormones; others are involved in innate immunity (which may be why epitheliopeptides (i.e., skin) in many animals expresses high levels of "neuropeptides" and also explains some well-known neuropeptides potent antimicrobial action. The same peptides can be both! Lastly, neuropeptides are used during development and body plan

7formation. From snails to flies to mammals, during development, dozens of “neuropeptides” are made and used during early development (before the neurons appear) with these chemical gradients helping to organize the developing embryo, and then the developmental expression stops and resumes in neurons. Is this why so many prohormones are well conserved? The animals examined may use them in this way, they make peptide toxins and may use the prohormones in alternative ways which would impact the claims of ‘neuropeptide’ evolution being neuron centric. While the staining demonstrates they are in neurons, it doesn’t address alternative well-conserved functions. The citations 14-17 and claims that neuropeptides are related to the ancestral nervous system may need qualification.

Author Rebuttal to Initial comments

Responses to comments from the reviewer #1 on manuscript #NATECOLEVOL-220115650-T 1-1. “Neuropeptides” are defined by structural criteria, both in computational and here in biochemical approaches. Their expression in neurons has to be demonstrated for each putative “neuro”peptide and the presence of a “neuro”peptide does not identify a cell as a neuron, in particular not in animals that are an outgroup to other animals with neurons. My impression is that this is not consequently observed here. In Fig 2a and b, NPWa and VWYa are shown in cells with a typical neural morphology. In Fig2c, FGLa, WTGa, RWFa and GVFa-positive cells are described as sensory and endocrine-like neurons. But beyond the expression of these peptides, what is the basis for assuming that these are neural cells? This classification then appears to be used to label cells in *Mnemiopsis* (a different ctenophore) as “Neuron (peptidergic)” in Ext Fig 7 and to collapse them into one group “neuron(peptidergic)” in Fig 3. The presence of the processing and secretion machinery does not provide independent support for this assumption, as I would think that this machinery needs to be present in any cell that expresses these peptides, whether they are neural or not. Synapse” and “gene expression” genes appear to be similar to other cells, in particular “epithelial”. The criteria for calling a cell neural need to be clearly stated in the main text. If it is the expression of a “neuro”peptide, the issue could be addressed by separating those cells/metacells for which the antibody stainings (here or in Sachkova et al.) have shown neural morphology from those for which only the presence of the “neuro”peptide is used for their classification. They could be labelled potential neural cells or just peptidergic cells. My apologies in case I didn’t follow the authors rationale correctly, but if so, it would be good to explain it in more detail.

According to your suggestions, in the revised manuscript, only cells that are confirmed to have at least one protrusion (neurite) will be termed as “neurons”, otherwise they will be clearly stated as “peptide-expressing cells” or “peptidergic cells”. The definitions of each peptide were also renamed accordingly. The amidated peptides identified in this study were classified as amidated short peptides or simply peptides in general, and those that were confirmed to be expressed in the “neurons” by immunostaining were distinguished as “neuropeptides”. This distinction was also reflected in the dotplot categories shown in Figure 3, Figure 4f-g, Extended Data Figure 6, and Supplementary Fig. 5 of the revised manuscript. For the above redefinitions, we carefully

reexamined the peptide staining using a spinning disc confocal microscopy etc. Our new findings, which we added to the revised manuscript, are described below.

- 1) In addition to NPWa and VWY_a, WTG_a was found to be expressed in cells with neurites in larvae of *B. mikado*. These WTG_a-positive neurons, as well as NPWa and VWY_a, are components of the SNN system.
- 2) In addition to NPWa, VWY_a and WTG_a develop the nerve nets at the base of comb rows.
- 3) In addition to VWY_a, WTG_a and RWF_a signal were also observed in the larval tentacles of *B. mikado*.
- 4) FGL_a was expressed also in cells of the pharynx.

The peptide expression patterns are consistent with that of their precursors (Sachkova et al.). With these new data, we define the NPWa, VWY_a and WTG_a as neuropeptides in the revised manuscript (**line 162**) "Immunostaining of *B. mikado* cydippid larvae using NPWa, VWY_a, and WTG_a antibodies visualized cells with cell processes (neurites), indicating that these amidated short peptides are functional in neurons". Because no clear neurites were observed in cells at the tentacles, pharynx and mouth, we labeled them simply as peptidergic or peptide-expressing cells. We also recategorized the cell clusters in the revised manuscript. For example, **line 204** "Among these, four cell clusters (C33, C34, C35, and C40) express the VWY_a, NPWa, and WTG_a neuropeptides. As the immunostaining confirmed that ...".

As can be seen from the above changes, we agree that the reviewer criterion "only cells with neurites are neurons" is currently the most reasonable one when based on our textbook view. However, we are concerned that the application of this criterion may overlook the neurophysiological properties that are shared with peptidergic cells and neurons, especially if we aim to understand the most ancestral neurons or their evolutionary predecessor(s). In addition, definitions based solely on observations at specific stages of development or in a very limited number of species may not provide the ideal criterion for understanding hidden plasticity that may be involved in the functional modalities of the primordial nervous systems.

Our detailed immunostaining analysis revealed a variety of characteristics and developmental dynamics of peptidergic cell morphologies. For example, NPWa, VWY_a and WTG_a neuropeptides are expressed also in cells without neurites in *B. mikado* cydippid larvae. In adult *V. multiformis*, no NPWa-positive cell bearing neurite-like protrusion was observed. In cnidarian *N. vectensis*, we observed cell morphological changes from neurite-less peptidergic mode to neurite-bearing neuronal mode during larval development. Therefore, we would like to keep discussions about difficulty to define neurons in the basal metazoans. In the revised manuscript, we included a discussion about this uncertainty. (**line 312**) "A series of immunostaining of short peptides demonstrated a high degree of variation in the morphology and localization..."

1-2 *In the same vein: why are Mnemiopsis cl 27 and 29 listed as “Neuron (peptidergic)” in the same Ext Fig 7? Epithelial cluster 18 and 19 seem to be very similar.*

The clusters classified as neurons (peptidergic) in the old manuscript or peptidergic in the revised manuscript are cell clusters showing amidated peptide expression among clusters classified as neurons (*N. vectensis*) or unknown (*M. leidy*) in the Sebe-Pedros et al. 2018 articles. The clusters annotated as non-neuronal cell types (*N. vectensis*) or known cell types (*M. leidy*) in these papers are not included here. We apologize for the misunderstanding caused by our ambiguous description. In the revised manuscript, we have clearly indicated this point. **(line 198)** “To further examine the nature of basal metazoan neurons at the molecular level ...”

2. *From line 205 on, the expression of transcription factors is used to show similarities to the regulatory program of bilaterian neurons. I didn’t understand how the TFs in this list were selected. Would there be paralogs (other bHLH or soxB genes) that are not in the “neuron(peptidergic)” clusters but enriched in other cell types? Since some of these TFs (e.g. sox1, BarHx, six1/2) are rather broadly expressed, I was wondering about the strength of this correlation and whether it would be possible to use different TFs to make a different point (e.g. the peptide expressing cells are muscle cells or epithelial cells).*

There are number of known transcription factors (TFs) involved in a series of neural differentiation steps in bilaterian animals. Some families have been suggested to be involved in neurogenesis in cnidarians. We used these points as a reference when making the list of TFs. At the same time, if there are multiple homologs in each family of those TFs (for details, please see supplementary figures), the homologs with the highest expression levels are designated as representatives. This is to refer to the potential magnitude of the effect of these genes on cell function rather than the cell type specificity of their expressions. This also applies to other genes such as peptide processing enzymes and synaptic proteins. Since the neurogenic and neural genes predated the neurons and these genes don’t show exclusive expression pattern in *M. leidy* (this is at least in part why the single cell transcriptome analysis were not able to define neurons), our approach is an alternative way to look at the similarity in gene composition of neurons between ctenophore and other animals. Interestingly, this approach highlighted not only genetic similarities between ctenophore and cnidarian/bilaterian peptidergic systems, but also substantial level of genetic overlaps between the peptidergic cells and other non-neural cells (categorized as secretory/gastrodermis in *N. vectensis* and digestive/epithelial in *M. leidy*), which prompted us to propose the digestive neuron theory. To clarify our gene selection criterion, we added sentences in the method section of the revised manuscript. **(line 767)** “Highly expressed homologs are more likely to have greater potential magnitude of...”. Nevertheless, there is still needed to experimentally verify the neurogenic function of these TFs. We believe that the list provides strong TF candidates and will support experimental identification of TFs that drive the gene expression to impart peptidergic nature to the cells in the future.

3. for Fig 2d, please describe better or use a cartoon to show where in the animals the imaged area is.

To improve the clarity, and according to the reviewer's suggestion, we have modified Figure 2 in the revised manuscript.

4. In line 163, stereocilia are mentioned. I cannot discern stereocilia or any other type of cell protrusion in Fig 2d. This is relevant for describing these cells as sensory in the figure legend. Please provide a better image or change the text to remove stereocilia/cilia. Please change line 437 (figure legend to putative sensory cell, even if you can demonstrate stereocilia.

We agree with the reviewer's concern. There is no functional and structural data enough to support our interpretation that the observed F-actin staining indicates the stereocilia. According to the reviewer's comment, we removed the notation "stereocilia" in the old manuscript and changed "sensory cells" to "putative sensory cells" in the revise manuscript.

5. Line 175: a "close" species: please provide a reference for ctenophore phylogeny. "Close" is very subjective, also given the moderate level of overlap in the neuropeptide repertoire.

According to the reviewer's suggestion, the revised manuscript includes the reference that objectively show the close phylogenetic relationship between *Bolinopsis mikado* and *Mnemiopsis leidyi*. In the revised manuscript, we clearly mentioned their relationships. (line 182) "This is not surprising because both *M. leidyi* and *B. mikado* are belonging...". We also added the reference: Christianson, L. M., Johnson, S. B., Schultz, D. T., & Haddock, S. H. D. (2022). Hidden diversity of Ctenophora revealed by new mitochondrial COI primers and sequences. *Molecular Ecology Resources*, 22, 283–294. <https://doi.org/10.1111/1755-0998.13459>.

6. Line 194: The labelling indeed seems to be not restricted to typical synaptic puncta. But I don't think there is homogenous distribution of the neuropeptide signal. "Broad" would be better.

We have now used "broad", instead of "homologous" in the revised manuscript, and the statement was changed to "Immunostaining of neuropeptides also visualized the broad intracellular distribution of peptide-containing vesicles," and highlighted in the revised manuscript (line 226).

7. Line 201-202: This seems to assume that synapses should have the same molecular composition

as those in bilaterians, which is one of the contentious points for the question about the similarity of the ctenophore neurons.

To make the point clear, we changed the statement as follows in the revised manuscript (**line 234**). “The broad spatial arrangement of neuropeptide⁺ vesicles in ctenophore neurons is reminiscent of neuroendocrine cells in *Bilateria*, suggesting that the peptidergic signaling in *Ctenophora* is mediated by volume transmission.”

8. Whenever possible (e.g.. line 199) please avoid using “ctenophore” and indicate the species.

We modified the text accordingly.

9. Line 231: The lack of labelling in Fig2 makes it a bit difficult to relate the expression of NPW to the observed effect of the peptide treatment. Is it possible to indicate the position of the mouth in Fig 2?

For *B. mikado*, we added schematic drawings in the revised Figure 2 and Extended Data Figure 4 indicating tissues/positions focused in the photos. For *N. vectensis* we added asterisks at the position of the mouth (Extended Data Figure 3). Thank you again for your support to improve our manuscript.

10. Maybe I missed it and it is in the supplement: A table for comparing the *Mnemiopsis* and *Bolinopsis* neuropeptides would be nice, even though the methods for their identification differ.

We made additional list as the Supplementary table 3 which presenting a comparison of the neuropeptide genes identified in this analysis (*B. mikado*) with those previously predicted (*M. leidy*). Description of the supplementary data in the manuscript was modified accordingly.

11. At what age/stage of *Bolinopsis* was mass spec performed?

The samples for peptidomics analysis were prepared from adults of *B. mikado*, *N. vectensis*, and *E. fluviatilis*. The use of adult specimens in peptidomics helped to ensure the sufficient amount of samples needed to increase the signal detection rates. Information about which stage of the animals was used is included in the method section of the revised manuscript (**line 624**).

12. Are there any controls for the antibodies? Like pre-adsorption with antigen? Or Western?

We share the reviewer's caution regarding antibody specificity. To minimize the non-specific signal in the immunostaining analysis, we used neither antisera nor IgG fraction, but used antibodies that were all affinity-purified using peptide antigens. In the case of analysis using

12antiserum or IgG fraction as the primary antibody, it is effective to confirm the signal specificity by pre-absorption using the antigen peptide, but in the case of affinity purified antibody, the pre-absorption is not needed. We didn't perform the western-blotting since the mature peptides are too small to be subjected to SDS-PAGE gel where they go to the dye front.

To examine that the IHC staining is not from non-specifically bound secondary antibody, we performed new staining with or without the anti-NPWa and anti-VWYa primary antibody. As shown in the supplementary figures 6 and 7, we didn't detect neuronal signals in the *B. mikado* and *V. multiformis* in the absence of the primary antibody. This is helpful in distinguishing signals given only by secondary antibodies from those from affinity-purified antibodies. Additionally, in cases where peptide precursor mRNA expression and peptide antibody staining have been confirmed, the staining patterns look consistent in *M. leidyi* and *B. mikado*, also suggesting that the antibodies are able to specifically recognize the target mature peptides. To examine the antibody specificities in *N. vectensis*, we tried to knockdown neuropeptide by the siRNA electroporation that we have recently established based on the method by Karabulut et al. (doi:10.1016/j.ydbio.2019.01.005). We were able to find siRNAs which decrease mRNA expression level of RFa, PRGa, QWa neuropeptides. Our IHC staining of neuropeptide-KD planula larvae showed a reduction of signal intensity, demonstrating that the antibodies actually recognize the target neuropeptides. The data is also included as the supplementary figure 8 in the revised manuscript.

13. What is the counterstain in Ext Fig 3?

We stained neuropeptides with DAPI. We apologize and clarified in the revised manuscript.

14. Do the acidic cleavage sites require different proteases? If so, is this included in the analysis of the *Mnemiopsis* clusters?

Thank you for asking this important question. Unfortunately, we don't have a list of putative proteases that are responsible for peptide cleavage at the acidic sites. Although peptides that are cleaved at the acidic amino acid site are known also in bilaterian animals (<https://doi.org/10.1021/pr100358b>), the enzymes responsible for the cleavage have not been identified.

15. *The more philosophical question: do the data really provide much support for a common origin of neurons? The peptides appear to have very little conservation compared to other animals (and even among ctenophores), so I am wondering if we would consider it support for common origin if ctenophores would use a monoamine that is not used in any other nervous system. No need to answer this, but I would argue that the data are interesting irrespective of their implications for answering a question that might be rooted mainly in semantics.*

13Thank you for your opinion. In research on the evolutionary origins of neurons, in my opinion, there are issues in the semantics about how we define neurons and their homology. In order to solve this impregnable evolutionary biology puzzles, it is important to retain diverse views and debates. However, I also think it is important to proceed with the discussion, if we can, from a solid point of view based on experimental verification.

For the multiple peptides identified in this paper, we focused not on the conservation of sequence for peptides with short length and fast mutation rate, but on the that of the gene repertoire required for cells to function as peptidergic. Even in *Bilateria*, there is a wide variety of neuropeptide sequences. There are not many peptide families for which an evolutionary link can be detected between *Protostomia* and *Deuterostomia*, for example (Extended Data Figure 2). For the peptidergic neuron-related genes, the cnidarian/bilaterian genetic signature was found to be shared with *Ctenophora*, albeit in a broader expression pattern. Together with data on the morphology of peptide-expressing neurons and their regulatory function in muscle contraction of *Ctenophora*, our findings do not seem to support that the peptidergic neurons of the *Ctenophora* evolved independently from the peptidergic nervous systems of *Cnidaria/Bilateria*.

The validity of inferring the homology of early divergent animal neurons based on the conservation of structure and function of the chemical transmitter ligands themselves is debatable. Chemical transmitters, including monoamines and their analogues, function widely in intercellular communications in plants and unicellular microorganisms, and the evolutionary origin of their functionality therefore predates the emergence of neurons. Although the genetic repertoire of chemical transmitter synthase enzymes is rich in the cnidarian genomes, these genes do not show clear enriched expression in *Nematostella* neurons (Supplementary Data 3). Much of what we know about the presence and function of chemical transmitters like monoamines in cnidarian neurons has been obtained by indirect methods using antibodies. I believe that the endogenous chemicals that function as neurotransmitters should be scrutinized by recent analytical techniques, not only in *Ctenophora* but also in *Cnidaria*.

Responses to comments from the reviewer #2 on manuscript #NATECOLEVOL-220115650-T

1. *The molecular identity of ctenophore neurons has been reported recently by Sachkova et al, 2021. Specifically, C33 cell cluster (from Sebe-Pedros et al, 2018) was identified as a neuron of the subepithelial neural net (SNN), C34 – neuron of the pharyngeal neural net, C35 - sensory neurons located in the gut and around the mouth. Additionally, C27, C30, C31, C32, and C40 cell clusters were identified as sensory cells of the aboral organ and C55 as a peptidergic tentacle cell. Thus, transcriptomes of several peptidergic neurons and sensory cells have been revealed already (please see Data S2 and Figure 5a in Sachkova et al, 2021). I think this needs to be discussed in the manuscript.*

According to the reviewer's comment, we added discussions in the revised manuscript (see below). Sachkova et al. and we have identified cell clusters that specifically express the peptide precursors and mature peptides, respectively. WISH and/or IHC have experimentally validated the spatial expression patterns of peptides, demonstrating the morphologies and their localization of peptide-expressing cells including the neurite-bearing SNN neurons. These are very important findings for understanding the characteristics of the nervous system of ctenophores. However, there is still a need for validation to interconnect each neuronal cluster with each staining pattern proposed by Sachkova et al. The idea that C33 constitutes an SNN deems based on a process of elimination through comparison of the expression patterns of the precursor genes, which have a broad expression profile, with other precursor genes showing narrower expression profiles. We still need to examine whether the cluster C33 is actually consist only of SNN neurons, by co-staining of the precursor genes with other genes "specific" to C33, for example. Another reason we are hesitating to go deep too much into the *M. leidyi* cell clusters proposed by Sebe-Pedros et al, 2018 is that the current metacell models have not yet been validated by experiment. It has been reported that accurate identification of detailed neuronal types is difficult in single cell transcriptome analysis in the absence of appropriate supplementary information dictating cell type characteristics (Northcutt Biol Sci 2019 [10.1073/pnas.1911413116](https://doi.org/10.1073/pnas.1911413116)). Additionally, a comparison of the current model of adult *M. leidyi* "neuronal" clusters with the single-cell transcriptome data of the larvae will help to characterize each (sub)type of neuronal cells. Therefore, we think that, in addition to the genes that Sachkova et al. and we identified, further data are needed for a detailed and solid discussion of the differences in experimentally uncharacterized neuronal (sub)types. We believe that the peptides are ideal markers for physiological characterization of still enigmatic ctenophore neuronal and neuroendocrine systems.

Based on the uncertainties mentioned above, in the revised manuscript, we included discussions of the cell clusters and some uncertainties need be addressed in the future study. **(line 330)** "The localization of some of the identified peptides reveled by immunostaining cannot be..".

2. Localisation of ctenophore neuropeptides by immunostaining worked really nicely. I think it would be worth to discuss conservation of neuropeptide expression patterns between B. mikado, V. multiformis and Mnemiopsis. WTGa corresponds to ML02212a from Mnemiopsis, its localisation to SNN and aboral organ has been shown by both ISH and immunostaining by Sachkova et al, 2021 (please see Figure 2). Additionally, localisation of VWYα (ML02736a) to SNN and FGLamide (ML30511a) to aboral organ have been revealed by ISH (Sachkova et al, 2021, Figure 2, Fig S5). It would be also interesting to discuss the conservation of neuronal architecture: peptidergic SNN and neurons beneath the comb rows and ciliated grooves were reported in Mnemiopsis as well.

However I have a concern regarding the specificity of antibodies used for Nematostella and NPWa antibody. Peptides used for immunisation are just 2 to 3 residues long, it may result in non-specific binding to several targets. This choice of short peptides needs to be explained.

According to the reviewer's suggestion, we included discussion about the similarity of staining patterns of mRNA and the mature peptides between *M. leidy* and *B. mikado*, respectively. **(line 179)** "The expression of VWY_a and WTG_a mature peptides in neurons at SNN...". Additionally, we added a schematic drawing (Fig.2x') which clearly showing the expression patterns of peptide precursor mRNAs in the revised manuscript.

We also discussed in the revised manuscript about the similarity and difference between *B. mikado* and the benthic *V. multiformis*. **(line 187)** "We confirmed that neural network of VWY_a⁺ neurons exists also in this benthic ctenophore...".

Concerning the specificity of antibodies raised by short antigen peptide, we agree with the reviewer's concern. For the following reasons, we usually chose the C-terminal region to create antibody for short amidated neuropeptides. Our mass spectrometry analysis experimentally confirmed that several prohormone precursors produce multiple mature peptides with different sequences. The sequence comparison detected partial conservation of the precursor homologs different species. In both cases, the sequence variations are observed mainly in the N-terminal region, but the C-terminal region, which contains the amidation site, tended to be conserved. By generating antibody against the conserved C-terminal structure, we can produce the pan-VWY_a antibody, for example. In addition, we use C-terminally amidated peptide as the antigen. This is an effective way to avoid non-specific binding of the peptide antibody to non-physiological (non-amidated) short peptides or to similar sequences within proteins (neither of which are amidated). Additionally, we chose the conserved C-terminal region to maximize the coverage of peptides that can be recognized by the antibody. To experimentally examine the antibody specificity, we carried out some additional experiments both in ctenophores (*B. mikado* and *V. multiformis*) and *N. vectensis* (also see our response to comment #8 from reviewer #1). Briefly, In the staining of ctenophores *B. mikado* and *V. multiformis*, we confirmed no signal in control staining with only the secondary antibody (the supplementary figure 6 and 7 in the revised manuscript). In *N. vectensis*, we confirmed the IHC signal are drastically decreased by neuropeptide gene knockdown. We added this data as the supplementary figure 8 in the revised manuscript.

3. Analysis of peptidergic cells' transcriptome (Fig 3): "The dot plots show the summed normalized expression of the highest-expression gene (Methods) from all the cell clusters of each cell category (bottom)." The rationale behind this approach should be explained better since the logic of summing up gene expression levels across several clusters is not clear. Because neuropeptide-positive clusters might have significantly different gene expression profiles (e.g., one gene may be upregulated in one cell type but not detectable in another one), this approach may create artificial hybrid cell types. For example, RIMS (ML017713a) and Munc13 (ML24335a) are enriched in C35 but were not detected in C40. Another example is Secretagoin (ML03617a) enriched in C33 but not detected in C29 and C30. This approach may also create an additional bias due to different number of cell clusters included into each category. For example, Elav (ML220720a), is highly expressed in

comb metacells (C49-C50) however after summing up across the categories it appears that total expression is higher in Neurons (10 clusters) and not in Comb (3 clusters).

We showed the Figure 3 in the old manuscript as an overview focusing on the functional category of cell clusters, not to create cell hybrids. However, we agree that this can be misleading to the reader. In order to avoid it, all dot plots in the revised manuscript now show the expression profiles for each gene across the individual cell clusters of *M. leidyi* or *N. vectensis*. Thank you very much for your helpful comment.

5. *“Digestive neuron” is an interesting hypothesis. I suggest to discuss it in comparison to other recently proposed hypotheses. For example, Moroz et al, 2021 proposed that neurons evolved multiple times from unrelated types of secretory cells. Further, the “divide-and-conquer model” by Burkhardt & Jekely, 2021 suggested that ctenophores have several neuronal cell types that probably evolved from the diverse secretory cells, and only one of them (sensory neurons around the ctenophore mouth, C35) is homologous to bilaterian neurons.*

According to the reviewer’s suggestion, we added sentences in the discussion section in the revised manuscript. **(line 343)** “It has been proposed that neurons evolved independently from secretory cell types multiple times,...”, and **(line 357)** “It has been assumed that the C35 cell cluster is..”.

6. *Synapses with translucent synaptic vesicles do exist in ctenophores (Hernandez-Nicaise 1973); translucent vesicles are normally filled with small molecule transmitters. The same neuron of the nerve net has both peptidergic and synaptic vesicles (Sachkova et al, 2021). I think these facts need to be considered when discussing pre- and postsynaptic machinery (Line 202: «the modest equipment of pre- and postsynaptic machinery, suggests that the peptidergic signaling in Ctenophora is mediated by volume transmission») and the role of glutamate (“Given that glutamate is secreted mainly from the non-neural cells in Ctenophora, the network-dependent functionality of the nervous system might be an evolutionarily secondary system”).*

According to the reviewer’s suggestion, we included statements concerning the observation of putative synaptic vesicles in *M. leidyi* in the revised manuscript.

(line 254) “The most abundant SLC17 homolog (ML011726a) in *M. leidyi* is expressed exclusively in digestive cell clusters, ...”.

7. *Line 151: “... if they exist, the concentration should be lower than the other basal metazoans, which is consistent with the reduced complexity of the peptide receptor repertoire in Porifera....” – this statement is not really clear. I think the reduced complexity of peptide receptor repertoire rather*

correlates with reduced number of peptides, not their concentration. Additionally, I would suggest to rephrase “peptide receptor” as “peptide receptor homologues” since it has not been shown experimentally that they indeed bind peptides.

We agree with the reviewer’s comment and removed the statement mentioning the complexity of peptide receptors in the revised manuscript. We rephrased the “peptide receptor” according to the reviewer’s suggestion (**line 292**) “The complex expression pattern of the GPCR peptide receptor candidates,…”.

8. Line 543 – 544: “The complete list of identified peptides is provided in Supplementary Table 1 and 2.” In addition to the list of peptides of interest provided in these tables, full list of proteins identified by mass spectrometry should be made available. An easily accessible .csv file with full search engine output for all the studied organisms (including the sponge) is critical for reproducibility of this research.

We totally agree that the deposition of raw data material is important for reproducibility of the research. That is the reason why we deposited the LC-MS raw data, direct and full output file search engines as well as the list of proteins (peptide precursors). Also, “full search engine output” actually contains many spectral and metadata (e. g. search setting, protein id, AA sequence, matched peaks and assigned fragments), therefore creating simple csv file covering “full search engine output” is not realistic. As reviewer #3 appraised on her/his comment, we properly uploaded all the raw and curated data including the ones that the reviewer requested. We believe deposition of all data in organized fashion to dedicated repository is more beneficial to research community rather than including plain csv file. We modified method to explain availability of data in detail. (**line 690**) “raw output of search engines showing unfiltered list of identified peptides with ..”

9. Line 687 – 688: “If structural variation is not observed among the peptides within the family, we selected the longest.” It would be nice to explain the rationale behind choosing the longest variant. It is possible that after precursor cleavage at specific sites, peptides undergo further cleavage by aminopeptidases and therefore the longest variant would represent an intermediate step in the maturation rather than a final product.

In mass spectrometry-based peptide identification, N-terminally truncated forms of C-terminally amidated peptides can be detected occasionally because of endogenous degradation or experimental artifact during sample preparation. It is thus difficult to conclude the short variants of peptides are endogenous or not. Certainly, it is meaningful to compare in future experiments how specific and active the detected variants are. In our study, we prioritized the possibility of obtaining the most information in the functional analysis of peptides and the prediction of their receptors. That is, we were worried that choosing short peptide variants could overlook information on the N-terminal side that might be physiologically important. The points we mentioned above are clearly stated in the

revised manuscript. (line 859). “we selected the longest to avoid artificial short versions of peptides which could occur during sample preparation”

10. Line 397: «small molecule fractions» - I guess authors meant peptide fractions.

We changed the sentence accordingly.

11. Recent preprint by Yañez-Guerra et al, 2021 reported 2 neuropeptide precursors (phonexin and nesphatin) conserved in sponges. I think it's worth discussing in connection to the fact that mass spectrometry did not identify these peptides.

Regarding nesphatin, as Yañez-Guerra et al indicated in the article, the mature peptides are not supposed to be amidated because it lacks C-terminally Glycine as amide donor in the precursors. Our workflow only identifies peptides with C-terminal amidation, therefore nesphatin peptides were not identifiable even if such peptides exist in the animals we examined. On the other hand, the structures of some of phonexin precursors (including *M. leidyi*, *A. queenslandica* and *O. carmella*) has Glycine flanked by C-termini of the expected mature peptides in their structure, whereas *N. vectensis* homolog does not. From this data, we expected that *B. mikado* and *E. fluvatilis* may have amidated form of phonexin peptide. We thus searched the precursor sequence in our transcriptome data of *B. mikado* and *E. fluvatilis* but were not able to find any potential phonexin homolog genes. Naturally, peptides cannot be identified if the precursor gene is not present in the transcriptome dataset used for peptide-to-spectrum matching, and this is probably why we don't see the phonexin peptide in our mass spectrometry analyses.

To address the point raised by the reviewer, we added sentences in the discussion section of the revised manuscript. (line 302) “On the other hand, it requires peptides at high concentration in sample and precursor genes need to be present...”.

Responses to comments from the reviewer #3 on manuscript #NATECOLEVOL-220115650-T

(1) As a minor point, they report that in addition to the well-known basic residue cleavages in prohormones, they observe acidic (D/E) residues are cleaved in many of their precursors. While this is likely correct, there have been past reports that under acidic storage conditions, aspartic acid and some other residues in peptides are cleaved, and fraction cleaved depending on the surrounding amino acid sequences. Could their observation of acidic cleavages, at least partially, be a sample preparation artifact as they stored their peptides in a formic acid solution?

Although small amount (1 %) of formic acid is contained in the extraction solution used, which was removed by a vacuum centrifuge immediately after peptide extraction. The peptide

19samples were stored dry rather than in solution. Therefore, acidic cleavage sites we identified are not considered an artifact. In addition to this technical point, the conserved tendency of the amino acid composition around the cleavage site suggests the presence of cleavage motifs preferred by a particular proteinase. This also suggests non-random cleavage, eliminating the possibility that the detected acidic cleavage site is simply due to formic acid treatment.

(2) While amidation is certainly a hallmark of bioactive neuropeptides (and toxins: see below), in most animals, only a fraction of bioactive neuropeptides are amidated; their search algorithm appears to require this. Any thoughts on what fraction of peptides this excluded? Similarly, did they find a recognizable PAM enzyme in all the organisms studied? Or was it the related PHM and PHL duo which combine to have the same function?

As the reviewers pointed out, not all neuropeptides are amidated. For example, chordates are known to have a relatively lower proportion of amidated peptides. However, since the majority of reported cnidarian neuropeptides are amidated (Takahashi, *Front. Endocrinol.*, 27 May 2020 | <https://doi.org/10.3389/fendo.2020.00339>), focusing on amidated peptides is certainly effective in confidently identifying new neuropeptides especially in poorly studied animals. This study is the first report of experimental identification of ctenophore peptides and have no information of non-amidated peptides. Therefore we cannot say specific proportion of amidated peptides in all short peptides. In the revised manuscript, we clarified our strategy focused on amidated peptides and their limitations. **(line 300)** “Unlike sequence-based prediction, mass spectrometry identifies structures of novel peptides....”

Regarding PAM enzyme, we carefully analyzed the structure and phylogeny of PAM genes of basal metazoans including species we used in this study. Metazoans, including *Ctenophora*, bear at least one PAM gene consisting of both PHM and PHL/PAL domains. In the revised manuscript, we added and highlighted the sentence below. **(line 219)** “Our structure and phylogenetic analyses of ctenophore PAM proteins demonstrated...”

The domain structure and molecular phylogenetic tree of the complete PAM genes are also added as the supplementary Figure 1 and 2 in the revised manuscript.

(3) Many “neurotransmitter” predate neurons but are still cell secreted factors; bacteria use dopamine and other catecholamines as quorum sensing molecules, both bacteria and plants use serotonin (it is a known auxin), they are growth factors in animals, etc. The statements made throughout assume neurotransmitters are from neurons. Perhaps these statements should be modified slightly in terms of their evolutionary wide-spread non-neuronal functions.

As the reviewer pointed out, the small molecules generally regarded as neurotransmitters in animals are present and functional also in non-neuronal, plant, and even non-metazoan signaling

systems (e.g. Roshchina, Adv. Exp. Med Biol. 874 (2016) DOI 10.1007/978-1-4419-5576-0_2). Although evolutionarily wide-spread function of “neurotransmitter” is an important topic, the critical question we focused in this study is what kind of signaling molecule(s) was functionally deployed in the primordial nervous system. We are not addressing the ancestral (unicellular) functions of neurotransmitter-related chemicals. Nevertheless, the point raised by the reviewer is helpful for some readers to have a wider evolutionary aspect of chemical neurotransmitters. In the revised manuscript, we added sentences in the discussion regarding the wide-spread non-neuronal and premetazoan functions of chemical transmitters. **(Line 346)** “An issue here is that the chemical substances which are generally called “neurotransmitters,” are widely used...”.

(4) The ancient roles of some of these molecules become important when thinking about prohormone products as neuropeptides / hormones. Yeast process and secrete peptides for mating. In animals, there are many alternative uses for peptides from prohormones. One is to create peptide toxins, with some reports claiming this was the initial function of some prohormones; others are involved in innate immunity (which may be why epitheliopeptides (i.e., skin) in many animals expresses high levels of “neuropeptides” and also explains some well-known neuropeptides potent antimicrobial action. The same peptides can be both!

Lastly, neuropeptides are used during development and body plan formation. From snails to flies to mammals, during development, dozens of “neuropeptides” are made and used during early development (before the neurons appear) with these chemical gradients helping to organize the developing embryo, and then the developmental expression stops and resumes in neurons.

Is this why so many prohormones are well conserved? The animals examined may use them in this way, they make peptide toxins and may use the prohormones in alternative ways which would impact the claims of ‘neuropeptide’ evolution being neuron centric. While the staining demonstrates they are in neurons, it doesn’t address alternative well-conserved functions. The citations 14-17 and claims that neuropeptides are related to the ancestral nervous system may need qualification.

Thank you for asking the important question. Peptides are actually wide-spread molecules sometime with pleiotropic functions. The experimental identification of peptides has made it possible to analyze the functions of these peptides in early-branching animal phyla. In this study, we investigated neuronal expression patterns of these peptides and their involvement in behavioral regulation in order to clarify the physiological properties of the still-enigmatic ctenophore nervous systems, which are essential for addressing the evolutionary origin of neurons. Understanding the non-neural functions (e.g. embryonic development) of these peptides is unfortunately beyond the scope of this study, but it provides important additional set of information to assume the evolutionary process of peptide themselves and their signal transduction systems. For that purpose, more comprehensive functional assays, other than typical functional assays for neurotransmitters as we demonstrated in this study, are required to examine whether these peptides have “*alternative well-*

conserved functions". Although we do not have any results to address such alternative functions, the list of neuropeptides we identified together with the target receptor prediction can be a useful resource to explore such evolutionarily conserved functions of neuropeptides.

Regarding the comment "*The citations 14-17 and claims that neuropeptides are related to the ancestral nervous system may need qualification.*", we changed references to recent reviews referring examples. Thank you very much for your comments.

Decision Letter, first revision:

Dear Hiroshi,

Thank you for submitting your revised manuscript "Common Features of Metazoan Peptidergic Neurons Support Their Single Origin" (NATECOLEVOL-220115650A). It has now been seen again by the original reviewers and their comments are below. The reviewers find that the paper has improved in revision, and therefore we'll be happy in principle to publish it in Nature Ecology & Evolution, pending minor revisions to satisfy the reviewers' final requests and to comply with our editorial and formatting guidelines.

Sincerely,

{redacted}

Reviewer #1 (Remarks to the Author):

The authors have thoughtfully and thoroughly addressed all my comments and revised the manuscript in a convincing manner. Thank you for taking the time to reply also to the less specific comment #15.

Reviewer #2 (Remarks to the Author):

22I am completely satisfied with all the responses and modifications implemented by authors. I think it is a manuscript of high importance and it should be published in Nature Ecology and Evolution.

I just noticed several typos:

Line 207: "neurite-baring" should be "neurite-bearing"

Line 294: "implys" should be "implies"

Figure 2X: "comb row neruon" should be "comb row neuron"

Line 693: the link to the Jpost database is missing (it was included in the previous version)

Reviewer #3 (Remarks to the Author):

The authors have done a good job in responding to the reviewer comments, and have strengthened and improved the presentation considerably.

Decision Letter, final checks:

Dear Dr. Watanabe,

Thank you for your patience as we've prepared the guidelines for final submission of your Nature Ecology & Evolution manuscript, "Common Features of Metazoan Peptidergic Neurons Support Their Single Origin" (NATECOLEVOL-220115650A). Please carefully follow the step-by-step instructions provided in the attached file, and add a response in each row of the table to indicate the changes that you have made. Please also check and comment on any additional marked-up edits we have proposed within the text. Ensuring that each point is addressed will help to ensure that your revised manuscript can be swiftly handed over to our production team.

****We would like to start working on your revised paper, with all of the requested files and forms, as soon as possible (preferably within two weeks). Please get in contact with us immediately if you anticipate it taking more than two weeks to submit these revised files.****

In recognition of the time and expertise our reviewers provide to Nature Ecology & Evolution's editorial process, we would like to formally acknowledge their contribution to the external peer review of your

23manuscript entitled "Common Features of Metazoan Peptidergic Neurons Support Their Single Origin". For those reviewers who give their assent, we will be publishing their names alongside the published article.

Nature Ecology & Evolution offers a Transparent Peer Review option for new original research manuscripts submitted after December 1st, 2019. As part of this initiative, we encourage our authors to support increased transparency into the peer review process by agreeing to have the reviewer comments, author rebuttal letters, and editorial decision letters published as a Supplementary item. When you submit your final files please clearly state in your cover letter whether or not you would like to participate in this initiative. Please note that failure to state your preference will result in delays in accepting your manuscript for publication.

Cover suggestions

As you prepare your final files we encourage you to consider whether you have any images or illustrations that may be appropriate for use on the cover of Nature Ecology & Evolution.

Nature Ecology & Evolution has now transitioned to a unified Rights Collection system which will allow our Author Services team to quickly and easily collect the rights and permissions required to publish your work. Approximately 10 days after your paper is formally accepted, you will receive an email in providing you with a link to complete the grant of rights. If your paper is eligible for Open Access, our Author Services team will also be in touch regarding any additional information that may be required to arrange payment for your article.

Please note that *Nature Ecology & Evolution* is a Transformative Journal (TJ). Authors may publish their research with us through the traditional subscription access route or make their paper immediately open access through payment of an article-processing charge (APC). Authors will not be required to make a final decision about access to their article until it has been accepted. [Find out more about Transformative Journals](https://www.springernature.com/gp/open-research/transformative-journals)

Authors may need to take specific actions to achieve [compliance](https://www.springernature.com/gp/open-research/funding/policy-compliance-faqs) with funder and institutional open access mandates. If your research is supported by a funder that requires immediate open access (e.g. according to [Plan S principles](https://www.springernature.com/gp/open-research/plan-s-compliance)) then you should select the gold OA route, and we will direct you to the compliant route where possible. For authors selecting the subscription publication route, the journal's standard licensing terms will need to be accepted, including [self-archiving-and-license-to-publish](https://www.nature.com/nature-portfolio/editorial-policies/self-archiving-and-license-to-publish). Those licensing terms will supersede any other terms that the author or any third party may assert apply to any version of the manuscript.

Please use the following link for uploading these materials:
{redacted}

Best regards,

{redacted}

Reviewer #1:

Remarks to the Author:

The authors have thoughtfully and thoroughly addressed all my comments and revised the manuscript in a convincing manner. Thank you for taking the time to reply also to the less specific comment #15.

Reviewer #2:

Remarks to the Author:

I am completely satisfied with all the responses and modifications implemented by authors. I think it is a manuscript of high importance and it should be published in Nature Ecology and Evolution.

I just noticed several typos:

25Line 207: "neurite-baring" should be "neurite-bearing"

Line 294: "implys" should be "implies"

Figure 2X: "comb row neruon" should be "comb row neuron"

Line 693: the link to the Jpost database is missing (it was included in the previous version)

Reviewer #3:

Remarks to the Author:

The authors have done a good job in responding to the reviewer comments, and have strengthened and improved the presentation considerably.

Final Decision Letter:

21st June 2022

Dear Hiroshi,

We are pleased to inform you that your Article entitled "Mass spectrometry of short peptides reveals common features of metazoan peptidergic neurons", has now been accepted for publication in Nature Ecology & Evolution.

Over the next few weeks, your paper will be copyedited to ensure that it conforms to Nature Ecology and Evolution style. Once your paper is typeset, you will receive an email with a link to choose the appropriate publishing options for your paper and our Author Services team will be in touch regarding any additional information that may be required

You will not receive your proofs until the publishing agreement has been received through our system

Due to the importance of these deadlines, we ask you please us know now whether you will be difficult to contact over the next month. If this is the case, we ask you provide us with the contact information (email, phone and fax) of someone who will be able to check the proofs on your behalf, and who will be available to address any last-minute problems . Once your paper has been scheduled for online publication, the Nature press office will be in touch to confirm the details.

Acceptance of your manuscript is conditional on all authors' agreement with our publication policies (see www.nature.com/authors/policies/index.html). In particular your manuscript must not be published elsewhere and there must be no announcement of the work to any media outlet until the publication date (the day on which it is uploaded onto our web site).

26Please note that *Nature Ecology & Evolution* is a Transformative Journal (TJ). Authors may publish their research with us through the traditional subscription access route or make their paper immediately open access through payment of an article-processing charge (APC). Authors will not be required to make a final decision about access to their article until it has been accepted. [Find out more about Transformative Journals](https://www.springernature.com/gp/open-research/transformative-journals)

Authors may need to take specific actions to achieve [compliance with funder and institutional open access mandates](https://www.springernature.com/gp/open-research/funding/policy-compliance-faqs). If your research is supported by a funder that requires immediate open access (e.g. according to [Plan S principles](https://www.springernature.com/gp/open-research/plan-s-compliance)) then you should select the gold OA route, and we will direct you to the compliant route where possible. For authors selecting the subscription publication route, the journal's standard licensing terms will need to be accepted, including [self-archiving and license to publish](https://www.nature.com/nature-portfolio/editorial-policies/self-archiving-and-license-to-publish). Those licensing terms will supersede any other terms that the author or any third party may assert apply to any version of the manuscript.

We welcome the submission of potential cover material (including a short caption of around 40 words) related to your manuscript; suggestions should be sent to Nature Ecology & Evolution as electronic files (the image should be 300 dpi at 210 x 297 mm in either TIFF or JPEG format). Please note that such pictures should be selected more for their aesthetic appeal than for their scientific content, and that colour images work better than black and white or grayscale images. Please do not try to design a cover with the Nature Ecology & Evolution logo etc., and please do not submit composites of images related to your work. I am sure you will understand that we cannot make any promise as to whether any of your suggestions might be selected for the cover of the journal.

You can now use a single sign-on for all your accounts, view the status of all your manuscript submissions and reviews, access usage statistics for your published articles and download a record of

27your refereeing activity for the Nature journals.

You can generate the link yourself when you receive your article DOI by entering it here: http://authors.springernature.com/share.

[REDACTED]

P.S. Click on the following link if you would like to recommend Nature Ecology & Evolution to your librarian <http://www.nature.com/subscriptions/recommend.html#forms>

** Visit the Springer Nature Editorial and Publishing website at www.springernature.com/editorial-and-publishing-jobs for more information about our career opportunities. If you have any questions please click here. **